# Ferritin Mitochondrial (FTMT)-Driven Mitochondrial Ferroptosis in Vascular Smooth Muscle Cells: A Role of NCOA4 in Atherosclerosis Pathogenesis and Modulation by Gualou–Xiebai

**DOI:** 10.3390/nu17233713

**Published:** 2025-11-26

**Authors:** Li Zhu, Jun Gao, Zijian Liu, An Zhou, Hongfei Wu

**Affiliations:** 1School of Pharmacy, Anhui University of Chinese Medicine, Hefei 230012, China; zhuli2026@163.com (L.Z.);; 2Anhui Province Key Laboratory of Bioactive Natural Products, Hefei 230012, China; anzhou@ahtcm.edu.cn

**Keywords:** atherosclerosis, vascular smooth muscle cells, NCOA4, mitochondrial damage, ferroptosis, Gualou–Xiebai herb pair

## Abstract

**Background/Objectives**: Atherosclerosis (AS)-related cardiovascular diseases are a major global health threat, with vascular smooth muscle cells (VSMCs) phenotypic switching, abnormal proliferation, and migration as key progression drivers. Nuclear receptor coactivator 4 (NCOA4), a core ferritinophagy mediator overexpressed in AS plaques, may promote VSMCs ferroptosis by perturbing mitochondrial iron metabolism and ROS homeostasis, but precise mechanisms remain unclear. The classic Chinese herbal combination “Gualou-Xiebai” (GLXB) has anti-AS effects, yet how it modulates NCOA4-mediated ferroptosis to inhibit VSMCs’ functions is unknown. This study addresses this gap to advance GLXB’s therapeutic potential and identify AS targets. **Methods**: An AS model was established in ApoE^−/−^ mice by 12-week high-fat diet feeding, with model validation confirmed via ultrasound monitoring and H&E staining. NCOA4 was genetically modulated (knockdown and overexpression) to assess its role in plaque formation and lipid deposition using H&E staining, aortic imaging, immunofluorescence, and Western blotting. In vitro, VSMCs were stimulated with ox-LDL to induce proliferation and migration. NCOA4 was silenced using siRNA to examine associated ferroptosis levels and molecular mechanisms. Protein interactions between NCOA4 and the mitochondrial iron storage protein FTMT were evaluated by Co-IP and GST pull-down assays, while mitochondrial ROS (mitoROS) levels were measured to explore functional relationships. The extent of ferroptosis and the underlying regulatory mechanisms were assessed following treatment with GLXB-containing serum or transfection with small interfering RNA targeting LOX-1 (si-LOX-1). **Results**: NCOA4 knockdown reduced aortic lipid deposition, plaque burden, VSMC proliferation/migration, and mitochondrial ferroptosis. NCOA4 bound and suppressed FTMT, inducing mitochondrial iron overload, ROS accumulation, membrane depolarization, and ferroptosis. Combining NCOA4 silencing with FTMT inhibition elevated mitoROS, confirming the axis’s role in iron homeostasis. GLXB attenuated VSMCs dysregulation in vivo and in vitro, an effect abrogated by LOX-1 overexpression. **Conclusions**: NCOA4 promotes AS by binding FTMT, disrupting mitochondrial iron homeostasis, and triggering VSMCs ferroptosis. GLXB inhibits LOX-1-mediated NCOA4 expression, mitigating ferroptosis and VSMCs dysregulation, supporting its potential as a targeted anti-AS therapy.

## 1. Introduction

Atherosclerosis (AS) is a type of vascular lesion characterized by inflammation, lipid metabolism disorders, damage to vascular endothelial cells, and the abnormal proliferation and migration of vascular smooth muscle cells (VSMCs) [1,2]. This condition serves as the pathological foundation for numerous cardiovascular diseases [3]. VSMCs constitute pivotal pathological drivers throughout AS, with dysregulated proliferation and migration representing a fundamental pathomechanism in atherosclerotic plaque development [4,5]. A notable feature of AS plaque development is the abnormal proliferation and migration of VSMCs from the media layer to the intima layer within aortic vessels, which leads to vascular lumen stenosis and aggravates the progression of AS [6,7]. Therefore, regulating the proliferation and migration of VSMCs is essential for enhancing therapeutic strategies aimed at treating AS.

Ferroptosis execution is mechanistically governed by the integrated dysregulation of redox homeostasis, iron metabolism, phospholipid peroxidation, and mitochondrial bioenergetics, pathologically hallmarked by the lethal accumulation of reactive oxygen species (ROS) and membrane-localized toxic lipid peroxidation derivatives [8,9]. Mitochondria serve as central hubs for cellular iron and oxidative metabolism, playing a crucial role in ROS production; their functional state is intricately linked to ferroptosis [10,11]. While ROS can function as signaling molecules, elevated levels are cytotoxic and may induce oxidative cell death [12]. Evidence indicates that alterations in mitochondrial morphology—such as mitochondrial fission and cristae expansion—are closely associated with ferroptosis, with several effective inhibitors targeting mitochondria directly [10,13]. Notably, mitochondrial ferritin (FTMT), a key guardian of mitochondrial iron homeostasis, functions by chelating free iron ions to preserve the redox balance [14]. This nuclear-encoded protein is specifically targeted to the mitochondrial matrix, where it forms a ferroxidase-active cage structure to store iron safely. In contrast to cytosolic ferritin, FTMT exhibits low basal expression that is upregulated under stress, pinpointing its role as a dedicated mitochondrial iron buffer [15,16]. Its primary protective mechanism lies in sequestering the reactive labile iron pool, which directly limits the substrate available for the Fenton reaction, thus preventing the generation of deleterious reactive oxygen species and preserving mitochondrial function [17]. However, our current understanding of the molecular regulatory mechanisms involving FTMT in ferroptosis remains limited.

Nuclear receptor coactivator 4 (NCOA4) serves as a selective cargo receptor that specifically recognizes ferritin in the cytoplasm [18]. It binds to LC3-PE (microtubule-associated protein 1 light chain 3 conjugated with phosphatidylethanolamine) on the autophagosome membrane, thereby facilitating the transport of the ferritin-NCOA4-LC3 complex to lysosomes for degradation [19]. This process ultimately releases free iron, which drives ferroptosis. The mammalian iron storage system comprises three key components: ferritin heavy chain (FTH), ferritin light chain (FTL), and FTMT. FTH and FTL together form a cytosolic ferritin complex composed of 24 subunits [20,21]. Previous research conducted by our group has demonstrated that NCOA4-mediated degradation of ferritin, a process referred to as ferritinophagy, enhances the breakdown of cytoplasmic ferritin within endothelial cells situated in atherosclerotic plaques [22]. Specifically, NCOA4 facilitates lysosome-dependent degradation of FTH in endothelial cells, leading to the accumulation of free iron and exacerbating ferroptosis. Although FTMT exhibits sequence similarity and functional overlap with FTH regarding iron homeostasis and redox balance, its role in mitochondrial iron metabolism and autophagy remains poorly understood. This suggests that NCOA4 may regulate VSMCs ferroptosis by targeting the mitochondrion-specific iron storage system FTMT.

Gualou–Xiebai (GLXB), a classical Chinese herbal formula first documented in Zhang Zhongjing’s Eastern Han Dynasty treatise Jin Gui Yao Lue, is composed of Gualou (*Trichosanthis Fructus*, TF) and Xiebai (*Allii Macrostemonis Bulbus*, AMB) in a 2:1 ratio [23]. Literature surveys indicate that over 120 classical and modern clinical formulations contain GLXB, including representative preparations such as Gualou–Xiebai Banxia Tang, Gualou–Xiebai Baijiu Tang, Zhishi Xiebai Guizhi Tang, and Danlou Tablets [24]. Modern pharmacological studies have demonstrated that GLXB-based formulations exhibit multiple bioactive effects, including antioxidative stress, anti-apoptosis, anti-inflammation, and cardiac function improvement [25,26]. Recent findings suggest that GLXB may alleviate mitochondrial damage and ferroptosis in aortic endothelial cells of ApoE^−/−^ mice. However, it remains unknown whether GLXB attenuates AS by inhibiting the aberrant proliferation and migration of VSMCs, specifically through the suppression of NCOA4-mediated ferroptosis. Although GLXB has been extensively used in traditional practice for AS with phlegm-stasis syndrome, a comprehensive understanding of its fundamental mechanisms of action is still lacking and requires in-depth exploration.

Given the established significance of ferroptosis in AS pathogenesis, this study investigated the role of NCOA4-mediated ferroptosis in AS and explored whether the classic Chinese medicine GLXB could attenuate AS by modulating this process. Using primary mouse VSMCs and AS mouse models, we demonstrate significant upregulation of both NCOA4 expression and ferroptosis activity within AS lesions. Notably, specific inhibition of NCOA4 promoted VSMCs’ proliferation and migration. Mechanistically, NCOA4 acts as a key regulator of mitochondrial iron homeostasis, directly interacting with FTMT to downregulate its expression. This NCOA4–FTMT interaction induces mitochondrial iron overload and aberrant ROS accumulation, culminating in ferroptosis. Crucially, the mitochondrial protection against ox-LDL insult conferred by NCOA4 knockout was significantly compromised under conditions of FTMT silencing. On this basis, we showed that GLXB administration ameliorated VSMC dysregulation. This protective effect was reversed by LOX-1 overexpression, confirming that GLXB’s action depends on the LOX-1/NCOA4 signaling pathway to inhibit ferroptosis and maintain VSMC homeostasis. Thus, our work establishes that GLXB targets LOX-1 to suppress mitochondrial ferroptosis, revealing its mechanistic value as a potential targeted therapy for AS.

## 2. Materials and Methods

### 2.1. Reagents

ox-LDL (YB001), Anti-NCOA4, Anti-GPX4, were obtained from Immunoway (YT0302, YN3047, Beijing, China). Anti-SLC7A11 (26864-1AP), Anti-GAPDH (TA-08). Gualou and Xiebai were purchased from Beijing Tong Ren Tang Co., Ltd. (Beijing, China; batch number: 211001, 211001). An ELISA kit was obtained from Jianglai Biotechnology (Shanghai, China). Erastin was purchased from SparkJade (SJ-MX0039, Jinan, China).

### 2.2. Animals

Ninety male ApoE^−/−^ mice and ten male wild-type C57BL/6J mice (6–8 weeks old, 20–22 g) were obtained from Jiangsu Hua Chuang Xin Nuo Pharmaceutical Technology Co., Ltd. (Taizhou, China). Following a one-week acclimatization period, the ApoE^−/−^ mice were randomly assigned to five experimental groups (*n* = 10 per group), while the C57BL/6J mice served as the genetic control group. Mice were maintained under specific pathogen-free (SPF) conditions in a controlled environment (temperature 22 ± 2 °C, humidity 50 ± 10%, 12 h light/12 h dark cycle). They were housed in standard ventilated cages (maximum 5 mice per cage) with corn cob bedding and provided ad libitum access to the respective experimental diets (normal chow or high-fat diet) and autoclaved drinking water. All procedures for animal experiments were approved by the Ethical Review Committee for Laboratory Animal Welfare of Anhui University of Chinese Medicine.

### 2.3. Animals Grouping and Treatment

C57BL/6 J mice (*n* = 10) were fed with a normal diet, and ApoE^−/−^ mice were fed with a high-fat diet (40% kcal fat, 20% kcal protein, 40% kcal carbohydrate, Cat # D12108C) to simulate an AS model. All animal experiments were approved by the Ethics Committee of Anhui University of Chinese Medicine (approval number: AHUCM-mouse-2023011, approval date: 15 October 2023). The animal experiments were conducted in accordance with the ethical standards established by the Laboratory Animal Welfare Committee of Anhui University of Traditional Chinese Medicine.

Mice were randomly assigned to groups and administered tail vein injections of the vehicle control (PBS with 0.1% BSA) at week 4 and week 8, and the observation period continued until week 16. At 16 weeks of age, mice were fasted for 12 h prior to euthanasia to minimize metabolic variability. Euthanasia was induced via intraperitoneal injection of sodium pentobarbital (50 mg/kg BW), a method approved by the Ethical Review Committee for Laboratory Animal Welfare of Anhui University of Chinese Medicine. Anesthesia depth was monitored by assessing the loss of corneal reflex and response to a painful toe pinch. Upon confirmation of absent vital signs for over 5 min, cervical dislocation was performed as a secondary step to ensure euthanasia. Immediately after confirmation of death, blood was collected via retro-orbital puncture, and tissues were harvested for subsequent analyses. Serum aliquots were stored at −80 °C without repeated freeze–thaw cycles following centrifugation; aortas were snap-frozen in liquid nitrogen and stored at −80 °C for molecular analyses or fixed in 4% paraformaldehyde for histological examinations. These procedures have been supplemented in the relevant Methods section to ensure reproducibility. The ApoE^−/−^ mice were randomly assigned to the following treatment groups (*n* = 10): (a) high-fat diet (HFD); (b) HFD + LvNCOA4; (c) HFD + LvNC; (d) HFD + shNCOA4; (e) HFD + shNC; GLXB-L (3 mg/kg) group, GLXB-M (6 mg/kg) group, GLXB-H (12 mg/kg) group, and simvastatin (5 mg/kg) group. The drugs were given once daily by oral gavage for 4 weeks. Finally, serum and aorta were collected for further analysis.

### 2.4. Lentiviral (Lv)-Infected ApoE^−/−^ Mice

To achieve NCOA4 overexpression and knockdown in atherosclerotic (AS) mice, we utilized lentiviral vectors from Gemma Biologicals (Shanghai, China), specifically Lv18E-NCOA4 (for overexpression) and NCOA4-MUS-1030/149/1345 (for knockdown), along with their respective controls. Lv18E-NCOA4 was constructed by cloning full-length mouse NCOA4 cDNA (NCBI GenBank accession no. NM_001145263.2) into the pCDH-CMV-MCS-EF1-copGFP lentiviral backbone, with the empty vector control (LvNC) using the same backbone without the NCOA4 insert; the genomic structure of the overexpression vector was CMV-MCS-hPGK-GFP-T2A-Puro. For knockdown, shNCOA4 and non-targeting scrambled control (shNC) were generated using the pLKO.1-puro lentiviral shRNA vector (Gemma Biologicals, catalog no. SHC001), with the shNCOA4 target sequence 5′-GCAACGUGAAGACCUACAATT-3′ and shNC showing no predicted homology to mouse transcripts. ApoE^−/−^ mice were administered 80 μL of each virus (titer: 1 × 10^9^ TU/mL) or their corresponding empty vectors via tail vein injection [22].

### 2.5. Preparation of GLXB Extract and In Vivo Administration

The GLXB herbal pair, composed of Gualou (Trichosanthes kirilowii Maxim) and Xiebai (Allium macrostemon Bunge), was procured from Beijing Tongrentang Co., Ltd. (Batch numbers: 211001, 211001, Beijing, China). The crude herbs at a weight ratio of 2:1 (Gualou:Xiebai; 100 g:50 g) were pulverized and subsequently macerated in five volumes of 50% aqueous ethanol for 1 h. Extraction was performed twice via reflux extraction, with each session lasting 2 h, using five volumes of 50% ethanol. The resulting extracts were pooled, concentrated to a final concentration of 0.6 g crude herb per milliliter, and stored at 4 °C [24]. All GLXB extracts were administered via oral gavage (intragastric administration) at a regimen of once daily, with consecutive administration for a total duration of 4 weeks.

### 2.6. Preparation of GLXB-Containing Serum

After one week of acclimatization, 20 Sprague Dawley rats were randomly assigned to a normal serum group (*n* = 10) and a GLXB serum group (*n* = 10). Rats in the GLXB group were orally administered the concentrated extract at a dose of 20 g/kg. One hour after the last administration, blood was collected from the abdominal aorta. The samples were incubated at 37 °C for 30 min and then centrifuged at 3000 rpm and 4 °C for 15 min. The supernatant was collected, filtered through a 0.22 μm membrane, and stored at −80 °C for subsequent experiments. The in vitro administration regimen was set as a co-incubation with 50 μg/mL ox-LDL and GLXB serum at a concentration of 24 h [22].

### 2.7. Cell Culture and Groupings

Mouse vascular smooth muscle cells were maintained in high-glucose DMEM supplemented with 10% FBS and 1% penicillin–streptomycin at 37 °C with 5% CO_2_. After serum starvation, VSMCs were divided into 12 groups. The control group was cultured in 10% DMEM, while the ox-LDL group was stimulated with 50 μg/mL ox-LDL for 24 h. The ox-LDL + Fer-1 group received 4 h pre-treatment with 1 μmol/L Fer-1 followed by 24 h co-incubation with ox-LDL. The ox-LDL + Erastin group was co-treated with 10 μmol/L Erastin and ox-LDL for 24 h. siRNA transfection groups (siNC, siNCOA4, siFTMT, and co-transfected siNCOA4 + siFTMT) were stimulated with ox-LDL 48 h post-transfection for 24 h. The ox-LDL + siNCOA4 + Erastin group was co-treated with Erastin and ox-LDL post-transfection. The ox-LDL + MitoQ group was pre-treated with 100 nmol/L MitoQ for 2 h before ox-LDL co-incubation. The MitoVES-containing groups (ox-LDL + MitoVES and ox-LDL + siNCOA4 + MitoVES) were treated with 5 μmol/L MitoVES and ox-LDL for 24 h. ox-LDL + GLXB-L (5%)/GLXB-M (10%)/GLXB-H (15%) groups (VSMCs stimulated with 50 μg/mL ox-LDL and cultured with 5, 10, and 15% GLXB-containing serum), ox-LDL + Lv-LOX-1 group (LOX-1 receptor overexpress VSMCs cultured with 50 μg/mL ox-LDL), ox-LDL+Lv-LOX-1 + GLXB-H group (LOX-1 receptor overexpress VSMCs cultured 15% GLXB-containing serum containing 50 μg/mL ox-LDL for 24 h [27].

### 2.8. Transfection of Lentivirus and Plasmid

After overnight incubation of VSMCs in a 6-well plate, the control plasmid was transfected using Lipofectamine 2000 (Solarbio, Beijing, China), 6 h post-transfection, the culture medium was replaced, and the cells were treated with ox-LDL for 24 h. Subsequently, VSMCs were transfected with either siNCOA4 constructs (siNCOA4-1, siNCOA4-2, siNCOA4-3) or an NCOA4 overexpression lentivirus (Lv18E; CMV-MCS-hPGK-GFP-T2A-Puro). A concentration of 5 μg/mL Polybrene was employed for the transfection of the Lv18E lentivirus [22].

### 2.9. Stereomicroscopic Imaging of Aortas

The mouse eyeball was dissected with scissors along the midline of the abdomen, and the aorta was bluntly separated and removed in normal saline. Saline was injected into the heart to flush the blood vessels to remove excess blood. Subsequently, the adipose tissue around the aorta was carefully removed using microscissors and fibrous tweezers. Finally, the lipid accumulation in the mouse aorta was detected and photographed by stereomicroscope (SteREO Discovery V20, Zeiss, Jena, Germany).

### 2.10. Hematoxylin-Eosin (HE) Staining

Aortic tissues were fixed, paraffin-embedded, and sectioned at 5 μm. Sections were deparaffinized in xylene and rehydrated through a graded ethanol series (100%, 95%, 80%, and 70%; 2 min each). After washing, staining was performed with hematoxylin (3–8 min), differentiated in 1% acid ethanol, blued in 0.6% ammonia, and counterstained with eosin (1–3 min). Sections were then dehydrated through a reverse ethanol series (70%, 80%, 95%, and 100%; 2 min each), cleared in xylene, and mounted for microscopic (BX53, Olympus, Tokyo, Japan) examination and ImageJ analysis.

### 2.11. Immunofluorescence Staining

After washing with PBS, the tissue sections were blocked for 2 h in PBS containing 0.3% Triton X-100 and 0.5% bovine serum albumin. The primary antibodies were prepared and incubated at 4 °C overnight. The secondary antibody was incubated for 2 h. DAPI was dropped and incubated in the dark for 15 min. PBS was added, and the washing was repeated 3 times. They were observed and photographed using a laser confocal microscope.

### 2.12. 5-Ethynyl-2′-Deoxyuridine (EdU) Analysis

Well-grown VSMCs were seeded in 6-well plates, and the reaction was continued for 1.5 h in the cell incubator after the addition of EdU working solution. After washing with PBS buffer, 1 mL of 4% PFA was added and fixed for 15 min at room temperature. An additional 1 mL of permeabilization solution (0.3% Triton X-100 in PBS) was added and incubated for 15 min at room temperature. A click reaction solution was prepared and reacted at room temperature in the dark for 30 min. DAPI was dropped and incubated in the dark for 15 min and washed with PBS. They were observed and photographed using a fluorescence microscope.

### 2.13. Migration Assay

Aortic vascular smooth muscle cells (VSMCs) were seeded at 1 × 10^6^ cells per well in 6-well plates and cultured until 100% confluence. A sterile 200 μL pipette tip was used to create uniform vertical scratches across the cell monolayer, and detached cells were removed by washing three times with PBS. Subsequently, cells were subjected to experimental treatments (consistent with in vitro treatment conditions described in Section 2.5). Images of the scratch areas were captured at 0 h and 24 h post-injury using an inverted phase-contrast microscope (Olympus Inc., Japan). The scratch healing rate was calculated as [(scratch width at 0 h − scratch width at 24 h)/scratch width at 0 h] × 100% using ImageJ software 1.52v.

### 2.14. Transwell Migration Assay

VSMCs were seeded at 1 × 10^4^ cells per well in the upper chamber of 24-well transwell inserts (8 μm pore size, Labgic, Beijing, China) containing serum-free DMEM. The lower chamber was filled with 600 μL complete DMEM (10% fetal bovine serum). The same three experimental treatments as above were applied to the upper chamber medium. After 24 h of incubation at 37 °C with 5% CO_2_, non-migrated cells on the upper surface of the membrane were gently removed with a cotton swab. Migrated cells adhering to the lower surface were fixed with 4% paraformaldehyde for 15 min, stained with 0.5% crystal violet solution for 10 min, and rinsed three times with PBS to remove excess dye. Images of migrated cells were acquired from five random fields per insert using an inverted phase-contrast microscope (Olympus Inc., Japan) at 100× magnification. The number of migrated cells was quantified using ImageJ software.

### 2.15. Western Blot Analysis

Total protein was extracted from aortic tissues (50 mg, minced on ice) and VSMCs using pre-cooled RIPA lysis buffer (P0013B, Beyotime, Shanghai, China) supplemented with 1× protease/phosphatase inhibitor cocktails. Tissue samples were homogenized (TissueLyser II, QIAGEN N.V., Venlo, The Netherlands) at 50 Hz for 2 min (twice, 30 s interval on ice) with 500 μL lysis buffer; VSMCs were rinsed twice with pre-cooled PBS, scraped in 200 μL lysis buffer per 6 wells, and vortexed gently for 10 min. All lysates were incubated on ice for 30 min and centrifuged at 12,000× *g* for 15 min (4 °C), and supernatants were collected. Protein concentration was determined via BCA assay (#23225, Thermo, Waltham, MA, USA). Equal protein amounts were separated by 10% SDS-PAGE, transferred to PVDF membranes (IPVH00010, Millipore, Darmstadt, Germany), blocked with 5% non-fat milk in TBST (1 h, RT), incubated with primary antibodies (1:1000, 4 °C, overnight) and HRP-conjugated secondary antibodies (1:5000, 1 h, RT), and visualized with ECL kit (Thermo Fisher, 34580) using an Ultra-Sensitive Multifunctional Imager (Cytiva, Marlborough, MA, USA) to measure the exposure, and band intensities quantified via ImageJ.

### 2.16. Co-Immunoprecipitation (Co-IP) Analysis

Protein samples were collected from treated cells, and protein concentrations were measured using the BCA Protein Assay kit (Pierce^TM^, #23225, Thermo, Waltham, MA, USA). Subsequently, 800 μg of protein in lysis buffer was immunoprecipitated using Protein A/G magnetic beads (DynaGreen^TM^, Thermo, Waltham, MA, USA). Equal volumes of each protein sample were separated by SDS-PAGE and transferred to PVDF membranes. After blocking with 5% (*w*/*v*) non-fat dry milk in TBST for 1 h at room temperature, the membranes were incubated with primary antibody overnight at 4 °C, followed by incubation with HRP-conjugated secondary antibody for 1 h at 25 °C. Protein signals were detected using a chemiluminescence imager [22].

### 2.17. GST Pull-Down Assay

Briefly, purified GST-tagged bait protein and His-tagged prey protein were incubated with glutathione-sepharose beads (Pierce^TM^, Thermo, Waltham, MA, USA) in binding buffer (50 mM Tris-HCl pH 7.5, 150 mM NaCl, 1 mM EDTA, 0.1% NP-40, 10% glycerol) at 4 °C for 2 h with gentle rotation. Beads were then washed five times with ice-cold wash buffer to remove non-specific bindings. Bound proteins were eluted by boiling in 2× SDS loading buffer and analyzed by SDS-PAGE followed by Western blotting using anti-His and anti-GST antibodies (abcam). Control groups with GST protein alone were included to confirm specific interactions.

### 2.18. Biochemical Analysis

VSMCs in logarithmic growth phase were seeded into 6-well plates at a density of 5 × 10^5^ cells/well and allowed to adhere for 24 h. Cells were then treated according to experimental groups with ox-LDL (50 μg/mL), Ferrostatin-1 (Fer-1, 1 μmol/L), Erastin (10 μmol/L), or their combinations, consistent with Section 2.7. After treatment, cells were washed with pre-cooled PBS, harvested by trypsinization, and centrifuged (1000× *g*, 5 min, 4 °C). The pellets were resuspended in 200 μL pre-cooled saline, homogenized on ice (30 s × 3 cycles), and centrifuged (3000× *g*, 10 min, 4 °C) to obtain supernatants. GSH, SOD, MDA, and Fe contents were determined using specific commercial kits (Jiancheng Bioengineering: A006-2-1, A001-3-2, A003-1-2; Abcam: ab83366, Nanjing, China) following manufacturers’ protocols, with three technical replicates and results normalized to protein concentration.

### 2.19. Transmission Electron Microscope (TEM) Analysis

One millimeter of fresh mouse aortic tissue was harvested and quickly frozen on ice. It was immediately placed in a 1.5 mL EP tube of electron microscope fixative solution and fixed at 4 °C in the dark for 24 h, then rinsed 3 times with PBS for 15 min each time. After fixation with osmium tetroxide for 2 h, the cells were rinsed 3 times with PBS for 15 min each time. They were then routinely dehydrated and embedded, stained with lead citrate and uranyl acetate (70–90 nm), and finally imaged with mitochondrial ultrastructure using transmission electron microscopy.

### 2.20. Flow Cytometry Analysis

VSMCs were seeded in 6-well plates, and the supernatant was discarded and washed 3 times with PBS. The prepared 1 μmol/L ROS working solution was added, mixed, and incubated in the cell incubator in the dark for 30 min. The plates were resuspended three times in PBS and analyzed by Flow cytometry, and data were analyzed using FlowJo_v10.8.1.

### 2.21. Statistical Analysis

All the data appeared as mean ± standard deviation, and SPSS19.0 software was used to perform statistical analysis. When the data were normally distributed and the variance was homogeneous, one-way analysis of variance (ANOVA) and Student–Newman–Keuls (SNK) test were used for comparison between groups. A *p*-value less than 0.05 was considered statistically significant.

## 3. Results

### 3.1. NCOA4-Mediated Ferroptosis Inhibits the Proliferation and Migration of VSMCs

We initially established models for gene silencing and overexpression of AS and NCOA4. Western blot analysis confirmed the successful establishment of an aorta-specific LvNCOA4 overexpression and shNCOA4 knockdown model in ApoE^−/−^ mice (Appendix A). Stereomicroscopy and HE staining further validated the promoting effect of NCOA4 in the AS model (Appendix A). Additionally, immunofluorescence analysis demonstrated that NCOA4 knockout significantly reduced the area occupied by VSMCs within AS plaques, whereas NCOA4 overexpression yielded the opposite effect (Figure 1A). This observation was corroborated by Western blot results, which indicated a significant upregulation of proliferation and migration markers PCNA and MMP2 in aortic VSMCs from the model group, alongside a notable downregulation of the contractile protein CNN1. In contrast, these effects were reversed in the shNCOA4 group while being exacerbated in the LvNCOA4 group (Figure 1B). These findings suggest that NCOA4 plays a critical role in promoting AS progression, consistent with previous research reports.

To further elucidate the mechanisms underlying the role of NCOA4 in AS progression, we assessed ferroptosis in VSMCs using TEM. TEM analysis revealed a significant reduction in mitochondrial swelling and cristae disruption in VSMCs with NCOA4 knockout, which are characteristic morphological features of ferroptosis (Figure 1C). Immunofluorescence colocalization demonstrated a significant decrease in the colocalization of α-SMA and GPX4 in the HFD group, an effect that was markedly attenuated by NCOA4 knockdown (Figure 1D, Appendix A). Western blot analysis further revealed that the expression of GPX4 and SLC7A11 was significantly increased in the aortas of NCOA4 knockout mice, whereas NCOA4 overexpression led to a decrease in anti-ferroptotic proteins in mouse aortic tissues (Figure 1E). Moreover, biochemical assays confirmed that NCOA4 knockdown inhibited the levels of ferroptosis markers LPO and MDA while upregulating the levels of the antioxidants GSH and SOD. Conversely, NCOA4 overexpression reversed these effects (Figure 1F–I). In summary, these findings demonstrate that NCOA4 promotes the proliferation and migration of VSMCs in AS mice by inducing ferroptosis.

### 3.2. NCOA4 Knockdown Reduces ox-LDL-Induced VSMCs Proliferation and Migration by Suppressing Ferroptosis

In vitro, we treated VSMCs with ox-LDL and downregulated NCOA4 expression using small interfering RNA (siRNA) (Appendix A). As illustrated in Figure 2A,B, silencing NCOA4 significantly reversed the abnormal proliferation and migration of VSMCs induced by ox-LDL. To investigate whether the effects of NCOA4 silencing on proliferation and migration were mediated through ferroptosis, we introduced the ferroptosis inhibitor Fer-1 and the ferroptosis inducer Erastin. Consistent results from EdU immunofluorescence staining as well as scratch and transwell assays indicated that Erastin treatment could counteract the effects of NCOA4 knockout on both proliferation and migration. Western blot analysis revealed that both Fer-1 treatment and NCOA4 silencing led to a downregulation of key markers associated with proliferation and migration, specifically PCNA and MMP2; however, these effects were reversed by Erastin treatment (Figure 2C). These findings suggest that silencing NCOA4 may inhibit the aberrant proliferation and migration of VSMCs in vitro by suppressing ferroptosis.

Notably, NCOA4 knockout resulted in a significant upregulation of protein expression for critical ferroptosis regulators SLC7A11 and GPX4—an effect that was antagonized by Erastin treatment (Figure 2D). In assays measuring oxidative stress markers, it was observed that NCOA4 silencing effectively mitigated the ox-LDL-induced reduction in levels of antioxidant enzymes such as SOD and GSH while also inhibiting the accumulation of lipid peroxidation products, including MDA and free iron ions (Appendix A). Further validation through fluorescence probe assays combined with flow cytometry demonstrated that NCOA4 knockout significantly reduced intracellular ROS levels within VSMCs—a phenomenon abolished upon Erastin treatment (Figure 2E,F). In summary, our results indicate that NCOA4 plays a pivotal role in promoting abnormal proliferation and migration of VSMCs via its mediation of ferroptosis.

### 3.3. NCOA4 Facilitates FTMT Degradation via Direct Interaction

Ferritin, composed of FTH heavy chains and FTL light chains, along with FTMT, constitutes essential intracellular iron storage systems [28]. As a selective autophagy receptor for ferritin, NCOA4 serves as a pivotal link between iron metabolism and ferroptosis [29]. To define the role of NCOA4 in iron homeostasis, we established an ox-LDL-stimulated VSMCs model. Notably, the knockout of NCOA4 significantly increased the expression levels of iron storage proteins FTH1 and FTMT without altering FTL levels (Figure 3A,B), consistent with previous reports. FTMT exhibits functional and structural homology with FTH, suggesting conserved roles in regulating the iron–redox balance. Quantitative immunoblotting confirmed that NCOA4 deficiency significantly reduced FTMT expression (Figure 3C,D), implicating NCOA4 in mitochondrial iron regulation. Molecular docking identified a stable NCOA4-FTMT complex stabilized by hydrogen bonds between key residues (Figure 3E). Critically, endogenous Co-IP validated specific NCOA4-FTMT interactions in VSMCs (Figure 3F), while GST pull-down assays using purified GST-NCOA4 demonstrated direct binding (Figure 3G). Immunofluorescence corroborated cytoplasmic colocalization of NCOA4 and FTMT (Figure 3H). These data establish that NCOA4 and FTMT physically interact in VSMCs under physiological conditions.

### 3.4. VSMCs-Specific FTMT Gene Knockout Enhances VSMCs Proliferation and Migration

To further elucidate the regulatory role of this interaction, we employed RNA interference technology to specifically silence the expression of the FTMT gene in VSMCs stimulated by ox-LDL (Appendix A). EdU staining analysis demonstrated that silencing FTMT significantly inhibited ox-LDL-induced proliferation of VSMCs (Figure 4A). Transwell migration assays and scratch wound healing experiments indicated that si-FTMT suppressed the enhanced migratory capacity induced by ox-LDL (Figure 4B), accompanied by a reduction in the expression levels of the proliferation marker PCNA and migration-related protein MMP2, while there was an upregulation of contraction phenotype marker CNN1 (Figure 4D). Notably, FTMT silencing markedly decreased intracellular reactive oxygen species (ROS) accumulation (Figure 4C) and reversed ferroptosis driven by ox-LDL, as evidenced by the restoration of GPX4 and SLC7A11 levels (Figure 4D). In summary, these data establish a crucial role for the NCOA4-FTMT interaction in coordinating phenotypic transitions in vascular smooth muscle cells. This is achieved through simultaneous regulation of iron storage dynamics via stabilization of FTMT and modulation of ferroptosis signaling pathways through GPX4/ACSL4 regulation. These findings provide mechanistic insights into how mitochondrial iron homeostasis governs vascular remodeling under conditions of oxidative stress.

### 3.5. NCOA4 Induces Mitochondrial Damage in VSMCs via FTMT Suppression and Iron Homeostasis Dysregulation

To further verify the role of NCOA4 in regulating mitochondrial damage through FTMT-mediated regulation of iron homeostasis within mitochondria, we established FTMT knockdown VSMCs and assessed the subsequent effects on mitochondrial function. The change in mitochondrial membrane potential showed that silencing NCOA4 increased the content of JC-1 aggregates and decreased the content of JC-1 monomer in VSMCs compared with the ox-LDL group, indicating that silencing NCOA4 alleviated ox-LDL-induced mitochondrial membrane potential reduction (Figure 5A). Furthermore, immunofluorescence staining showed that silencing NCOA4 significantly reduced the increase in Mito-Fe^2+^ content in VSMCs induced by ox-LDL. However, we observed that silencing FTMT reversed the therapeutic effects of silencing NCOA4. Dysfunctional iron homeostasis in mitochondria is an important cause of ROS accumulation in mitochondria [30]. Therefore, fluorescence microscopy was used to further investigate the ROS production in mitochondria. We found that silencing NCOA4 again reversed the ox-LDL-induced increase in MitoROS (Figure 5B). Similarly, the content of MitoROS in VSMCs increased again after silencing FTMT (Figure 5C). Taken together, these findings strongly suggest that silencing NCOA4 can regulate mitochondrial iron homeostasis by upregulating FTMT, thereby inhibiting the production of mitochondrial ROS and improving mitochondrial damage.

### 3.6. Silencing NCOA4 Inhibits Ferroptosis in VSMCs via Mitochondrial Protection

Mitochondria are the primary site of iron utilization and a key regulator of oxidative metabolism, serving as the principal source of ROS and a crucial locus for ferroptosis [31]. To explore whether silencing NCOA4 inhibits ferroptosis in VSMCs by improving mitochondrial damage, we examined the effect of NCOA4 on mitochondrial integrity and the impact of modulating mitochondrial ROS on ferroptosis. MitoQ, a mitochondrial ROS scavenger, and MitoVES, a mitochondrial ROS inducer, were employed to manipulate MitoROS levels [32,33]. As shown in Figure 6A, silencing NCOA4 significantly reduced ox-LDL-induced mitochondrial damage in VSMCs. In addition, MitoQ was used to inhibit MitoROS production, which could alleviate the abnormal reduction in mitochondrial membrane potential, whereas MitoVES treatment reversed the silencing of NCOA4, consequently exacerbating the impairment of mitochondrial membrane potential. These results indicated that silencing NCOA4 could alleviate mitochondrial damage by inhibiting MitoROS production.

To investigate the role of NCOA4 in modulating VSMCs ferroptosis via mitochondrial damage, we used MitoROS scavengers and inducers to modulate mitochondrial damage and examined their effects on ferroptosis levels in VSMCs. MitoQ was used to inhibit mitochondrial damage, and it reversed ox-LDL-induced decreases in GSH and SOD levels and increases in MDA and LPO levels in VSMCs (Figure 6B–E). Flow cytometry was used to detect the level of ROS in VSMCs, which showed that inhibition of mitochondrial damage reversed the ox-LDL-induced increase in ROS content in VSMCs. (Figure 6F,G). In addition, inhibition of mitochondrial damage significantly increased the protein expression of SLC7A11 and GPX4 compared with the ox-LDL group (Figure 6H). However, the opposite result was observed when MitoVES treatment aggravated mitochondrial damage and MitoVES reversed the therapeutic effect of silencing NCOA4. These findings indicate that mitochondrial damage is positively correlated with ferroptosis in VSMCs. Furthermore, silencing NCOA4 was confirmed to inhibit ferroptosis in VSMCs by reducing mitochondrial damage.

### 3.7. Silencing NCOA4 Inhibits Proliferation and Migration in VSMCs by Improving Mitochondrial Damage

Next, we explored whether NCOA4-induced mitochondrial damage would stimulate the proliferation and migration of VSMCs. We used EdU assays, scratch assays, and transwell assays to assess VSMCs’ proliferation and migration. As shown in Figure 7A,B, treatment with MitoQ significantly reduced EdU fluorescence levels, relative migration rates, and the number of migrating VSMCs compared to the ox-LDL group. Consistent with these results, MitoQ down-regulated the protein expression of PCNA and MMP2 while upregulating CNN1 expression (Figure 7C), indicating that alleviating mitochondrial damage can mitigate abnormal proliferation and migration in VSMCs. Conversely, treatment of VSMCs with MitoVES produced opposite effects. MitoVES induces mitochondrial damage and reverses the inhibitory effect of silencing NCOA4 on abnormal proliferation and migration of VSMCs. Collectively, these results demonstrate that silencing NCOA4 can alleviate ferroptosis and improve VSMCs proliferation and migration by inhibiting mitochondrial damage.

### 3.8. GLXB Attenuates Atherosclerotic Plaque Formation via NCOA4-Dependent Suppression of Ferroptosis in VSMCs

GLXB has a well-documented history in treating AS-related diseases, and our research team confirmed its strong anti-atherosclerotic effects [24]. Assessment of aortic plaque formation through enface photography and HE staining showed that GLXB significantly reduced lipid accumulation and the area of atherosclerotic lesions in AS mice compared to the HFD group in a dose-dependent manner (Figure 8A–C). To further explore GLXB’s impact on VSMCs proliferation and migration within the AS model, we conducted in vitro studies. EdU incorporation assays, wound healing assays, and transwell migration assays revealed that GLXB treatment increased EdU-positive cells compared to the ox-LDL group—indicating reduced inhibition of proliferation—and significantly suppressed both VSMCs migration rate and number (Figure 8D–H). Consistent with these findings, Western blot analysis indicated that GLXB downregulated proliferation marker PCNA and migration-associated protein MMP2 while upregulating contractile phenotype marker CNN1 relative to the ox-LDL group (Figure 8I).

To investigate the role of NCOA4 in the mechanism of GLXB, we assessed its expression levels in aortic plaques. Immunofluorescence staining revealed that NCOA4 levels were significantly elevated in the HFD group compared to the control group; importantly, GLXB treatment was found to dose-dependently suppress this increase (Figure 8J,K). Given the established link between NCOA4 and ferroptosis, we hypothesized that GLXB’s inhibition of plaque formation might involve modulating ferroptosis in VSMCs. To assess ferroptosis within aortic lesions, we performed co-staining for the VSMCs marker α-SMA and the key ferroptosis inhibitor GPX4. This analysis showed that GLXB significantly increased GPX4 protein expression compared to the HFD group (Appendix A). Further supporting this, biochemical assays revealed that GLXB treatment effectively lowered the excessive accumulation of MDA and LPO in the aortas of AS mice, while simultaneously boosting levels of reduced GSH and SOD activity (Figure 8L–O). Consistent with these findings, TEM demonstrated that GLXB treatment improved mitochondrial integrity in aortic VSMCs. Specifically, GLXB reduced mitochondrial cristae rupture, decreased the number of vacuolated mitochondria, and attenuated abnormal mitochondrial swelling and shrinkage, resulting in a restoration of near-normal mitochondrial ultrastructure (Figure 8P).

### 3.9. GLXB-Containing Serum Attenuates VSMCs Proliferation and Migration by Targeting LOX-1 to Suppress NCOA4-Dependent Ferroptosis

Our previous study demonstrated that GLXB attenuates AS by suppressing LOX-1-dependent cGAS-STING signaling, thereby inhibiting NCOA4-mediated ferroptosis in aortic tissue [22]. To investigate whether the LOX-1 receptor mediates GLXB’s suppression of VSMCs proliferation and migration, we first assessed its protein expression. As illustrated in Appendix A, ox-LDL stimulation significantly upregulated LOX-1 expression; however, treatment with GLXB dose-dependently reversed this induction. Subsequently, we examined the functional role of LOX-1 in mediating the effects of GLXB. EdU incorporation assays, scratch wound healing assays, and transwell assays revealed that overexpression of LOX-1 counteracted the inhibitory effects of GLXB on VSMCs proliferation and migration. Specifically, LOX-1 overexpression resulted in an increased area/number of EdU cells (Figure 9A,B) and accelerated wound closure (Figure 9C–E), indicating exacerbated dysregulation of VSMCs.

To assess mitochondrial involvement, we performed JC-1 staining. The results indicated that GLXB significantly mitigated ox-LDL-induced hyperpolarization of mitochondrial membrane potential (ΔΨm) (Figure 9F,H), suggesting protective effects against mitochondrial dysfunction. To establish a connection to ferroptosis, we measured key biomarkers associated with oxidative stress and lipid peroxidation. Notably, GLXB reversed ox-LDL-induced depletion of SOD and GSH while also suppressing accumulations of MDA and total iron levels in VSMCs (Figure 9J–M). Consistent with these findings, treatment with GLXB normalized elevated ROS levels induced by ox-LDL (Figure 9G,I). Mechanistically, Western blot analysis confirmed that GLXB upregulated proteins involved in ferroptosis defense mechanisms—specifically SLC7A11 and GPX4—while downregulating NCOA4 expression. Importantly, these protective effects were abolished upon overexpression of LOX-1 (Figure 9N,O). In summary, our data collectively demonstrate that GLXB mitigates mitochondrial damage and lipid peroxidation within VSMCs through targeting LOX-1; this action inhibits ferroptosis and exerts anti-atherosclerotic effects.

## 4. Discussion

AS is a chronic vascular disease characterized by the pathological proliferation and migration of VSMCs, involving multiple mechanisms. Ferroptosis significantly contributes to AS progression, with NCOA4 acting as a key regulator of intracellular iron homeostasis [22,34]. Our findings indicate that NCOA4 disrupts mitochondrial iron balance by regulating FTMT degradation, leading to mitochondrial damage and ferroptosis. We demonstrate a direct interaction between NCOA4 and FTMT; silencing FTMT reverses the effects of NCOA4 on mitochondrial function. In vivo experiments show that knocking out NCOA4 reduces AS plaque formation, VSMCs proliferation and migration, and markers associated with ferroptosis, while overexpression worsens the condition. These results highlight the essential role of NCOA4 in modulating VSMCs function and ferroptosis. Furthermore, our study suggests that combining Lv-NCOA4 with shNCOA4 gene knockout techniques can effectively establish an AS model. Our observations may lead to new therapeutic strategies for regulating VSMCs’ phenotypic changes and provide insights into the underlying mechanisms of AS.

Our research elucidates the pivotal role of NCOA4 in the progression of AS, particularly its mechanism for regulating VSMCs function through ferroptosis pathways. Prolonged exposure of VSMCs to a high-lipid environment prompts their transition from a contractile phenotype to a synthetic phenotype, thereby driving intimal hyperplasia and plaque formation, which are central to disease progression [2,35]. We observed that NCOA4 expression is significantly upregulated in the aortas of ApoE^−/−^ mice subjected to high-fat diets, as well as in VSMCs stimulated with ox-LDL in vitro. NCOA4 is crucial for regulating intracellular iron homeostasis. It promotes the degradation of FTH1, releasing free iron ions and directly participating in ferroptosis-related metabolic pathways [36,37,38,39]. Utilizing tail vein injection for lentiviral transfection, we established models of NCOA4 knockout and overexpression in VSMCs. The results demonstrated that silencing NCOA4 not only significantly inhibited the accumulation of lipid peroxidation products (such as MDA and LPO) and ROS induced by ox-LDL but also effectively suppressed VSMCs proliferation and migration. Furthermore, treatment with the ferroptosis inducer Erastin on silenced NCOA4 VSMCs markedly reversed these inhibitory effects. Importantly, knockout of NCOA4 significantly alleviated AS-related proliferation and migration of VSMCs induced by high-fat diets. These findings indicate that NCOA4 substantially influences VSMCs proliferation and migration in AS through modulation of ferroptosis processes; however, further investigation is warranted to elucidate the underlying mechanisms by which NCOA4 mediates ferroptosis in VSMCs.

Mitochondria serve as the central hub for oxidative metabolism in the body, acting as a primary source of ROS and being the main site for ferroptosis [40]. The accumulation of ROS exacerbated by mitochondrial iron overload is a key mechanism underlying oxidative-stress-induced ferroptosis in cardiomyocytes [41,42]. FTMT plays a crucial role in maintaining mitochondrial iron homeostasis; mice deficient in FTMT exhibit typical characteristics of ferroptosis [43]. As a critical iron storage protein within the organism, FTMT shares high sequence homology with FTH1 [21]. Our previous research has demonstrated that NCOA4 can induce ferroptosis in endothelial cells through ferritinophagy [22]. However, the regulatory mechanisms governing FTMT remain incompletely understood. In this study, we found that silencing NCOA4 directly upregulates FTMT protein expression, indicating that NCOA4 plays an important regulatory role in maintaining mitochondrial iron homeostasis. Furthermore, silencing NCOA4 alleviates mitochondrial iron overload and reduces ROS production by upregulating FTMT. This further confirms the reciprocal relationship between mitochondrial injury and ferroptosis. These findings suggest that NCOA4 not only induces ferroptosis through autophagic degradation of FTH1 but also promotes degradation of FTMT to induce mitochondrial injury, thereby facilitating ferroptosis. This study elucidates potential mechanisms by which NCOA4 regulates mitochondrial injury. Therefore, subsequent research focused on the role of NCOA4-mediated mitochondrial injury in ferroptosis of VSMCs. To address this issue, we employed MitoQ and Mitoves to modulate mitochondrial injury and demonstrated that such damage induces ferroptosis in VSMCs. Furthermore, the study revealed that mitochondrial injury exacerbates the proliferation and migration of VSMCs induced by ox-LDL. Notably, silencing NCOA4 alleviated mitochondrial damage, thereby improving the occurrence of ferroptosis as well as proliferation and migration in VSMCs. Thus, we have established that silencing NCOA4 can mitigate ferroptosis by upregulating FTMT expression to inhibit mitochondrial injury, ultimately improving the proliferation and migration of VSMCs.

GLXB, a traditional herbal pair, is clinically employed for treating chest obstruction syndrome attributed to phlegm-dampness blockage [44]. Contemporary pharmacological evidence indicates its utility in ameliorating hyperlipidemia, mitigating vascular inflammation, reducing aortic plaque formation, and managing AS-related pathologies [45]. In this study, we investigated the therapeutic potential of GLXB in ApoE^−/−^ mice with AS. Our results demonstrate that GLXB significantly attenuates atherosclerotic plaque development, diminishes lipid deposition within the aortic wall, and alleviates luminal stenosis. Notably, GLXB downregulated the expression of NCOA4 in VSMCs of AS mice. Preliminary investigations further revealed that GLXB treatment reduced the levels of proliferation marker PCNA and matrix metalloproteinase MMP2 while enhancing the expression of the contractile marker CNN1 in VSMCs. Importantly, we observed that GLXB upregulates key negative regulators of ferroptosis (GPX4 and SLC7A11) in aortic VSMCs, thereby attenuating lipid peroxidation and suppressing mitochondrial injury. Building upon these findings, we sought to elucidate the precise mechanisms through which GLXB modulates VSMCs’ proliferation and migration. We identified lectin-like LOX-1 as a critical mediator of GLXB’s therapeutic effects. GLXB inhibited aberrant VSMCs proliferation, migration, and ferroptosis primarily via the LOX-1 pathway, as evidenced by the reversal of its beneficial effects upon LOX-1 overexpression in vitro. Moreover, GLXB-mediated suppression of NCOA4 expression was also dependent on LOX-1 receptor signaling.

## 5. Conclusions

In conclusion, this study elucidates that NCOA4 directly binds to and downregulates FTMT expression, thereby disrupting mitochondrial iron homeostasis. This disruption leads to mitochondrial iron overload and aberrant accumulation of reactive oxygen species, ultimately driving the ferroptotic process in VSMCs. Conversely, the classic Chinese herbal medicine GLXB, by targeting the LOX-1 receptor, inhibits the NCOA4 signaling pathway, effectively attenuates mitochondrial ferroptosis, and maintains VSMCs homeostasis, thereby mitigating AS. These findings not only reveal the molecular mechanism by which the NCOA4-FTMT axis regulates mitochondrial ferroptosis in AS but also provide a theoretical foundation for GLXB as a potential targeted therapeutic strategy. While this study establishes the functional and physical interaction between NCOA4 and FTMT, the precise molecular pathway mediating FTMT degradation remains to be elucidated. We demonstrate NCOA4-dependent FTMT downregulation; however, whether this process relies on the canonical autophagy–lysosome pathway (e.g., mitophagy, given FTMT’s mitochondrial localization) or the proteasomal degradation system requires further validation via pathway-specific inhibitors (e.g., bafilomycin A1, MG132) or genetic ablation experiments in future studies.

## Figures and Tables

**Figure 1 nutrients-17-03713-f001:**
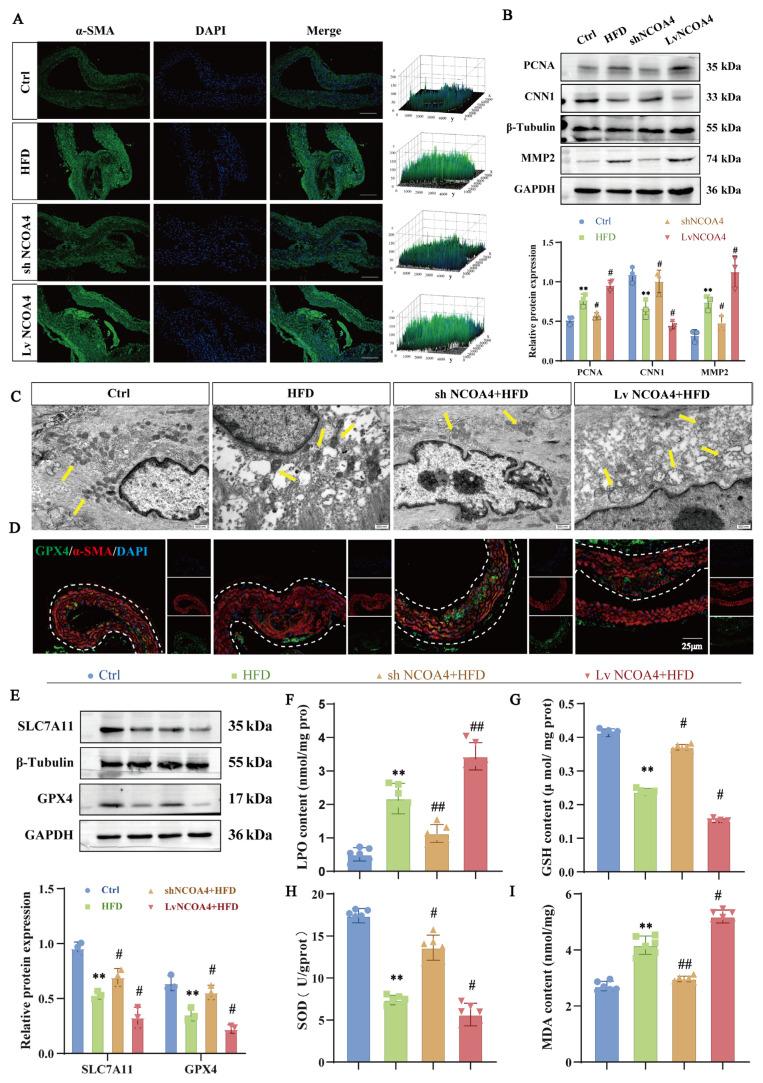
NCOA4-mediated ferroptosis inhibits the proliferation and migration of VSMCs. (**A**) Immunofluorescence staining of α-SMA (green) and DAPI (blue) in mouse aorta (scale bar: 100 µm, *n* = 3). (**B**) Western blotting to detect the expression of proliferation and migration proteins PCNA, MMP2 and CNN1 in the aorta of AS mice (*n* = 3). (**C**) TEM to detect the mitochondrial injury in the aortic VSMCs of AS mice (yellow arrows indicate mitochondria; scale bar: 500 nm, *n* = 3). (**D**) Immunofluorescence staining of α-SMA (red) GPX4 (green) is co-localized in sections of atherosclerotic lesions (scale bar: 25 µm, *n* = 3). (**E**) Western blotting to detect the expression of ferroptosis proteins GPX4 and SLC7A11 in the aorta of AS mice (*n* = 3). (**F**–**I**) Biochemical kits to detect the expression of lipid peroxidation levels of MDA, LPO, GSH, and SOD in the aortic tissues of AS mice (*n* = 6). ** *p* < 0.01 vs. control group; ^#^
*p* < 0.05, ^##^
*p* < 0.01 vs. HFD group.

**Figure 2 nutrients-17-03713-f002:**
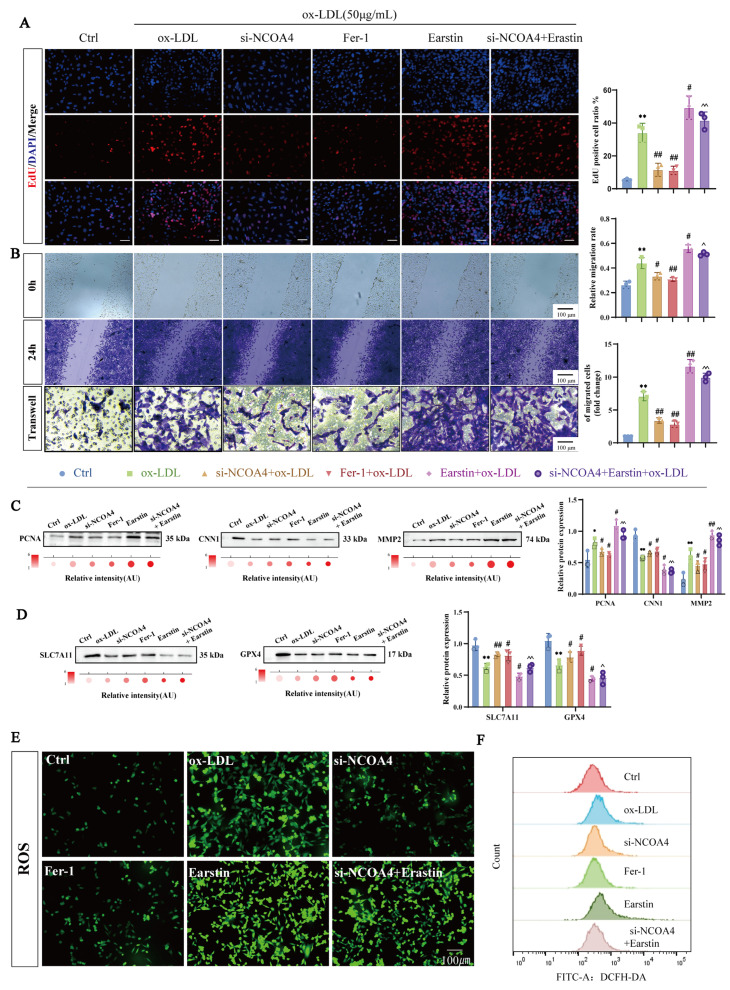
NCOA4 knockdown inhibits ferroptosis and reduces ox-LDL-induced VSMCs proliferation and migration. (**A**) Proliferation analysis in mouse VSMCs using fluorescent EdU assay (red); DAPI was used to stain nuclei (blue) (scale bar: 50 µm, *n* = 3). (**B**) Evaluation of VSMCs migration using transwell assay and wound healing assay (scratch assay) in mouse VSMCs. The gap created by scratch at two time-points (0 h and 24 h) and the migration of VSMCs was calculated (scale bar: 100 µm, *n* = 3). (**C**,**D**) Western blotting to detect the expression of PCNA, MMP2, CNN1, and ferroptosis-related protein in VSMCs. (**E**) Immunofluorescence to detect intracellular ROS level (*n* = 3). (**F**) Flow cytometry for ROS levels (scale bar: 100 µm, *n* = 3). * *p* < 0.05, ** *p* < 0.01 vs. control group, ^#^
*p* < 0.05, ^##^
*p* < 0.01 vs. ox-LDL group, ^ *p* < 0.05, ^^ *p* < 0.01 vs. si-NCOA4+ox-LDL group.

**Figure 3 nutrients-17-03713-f003:**
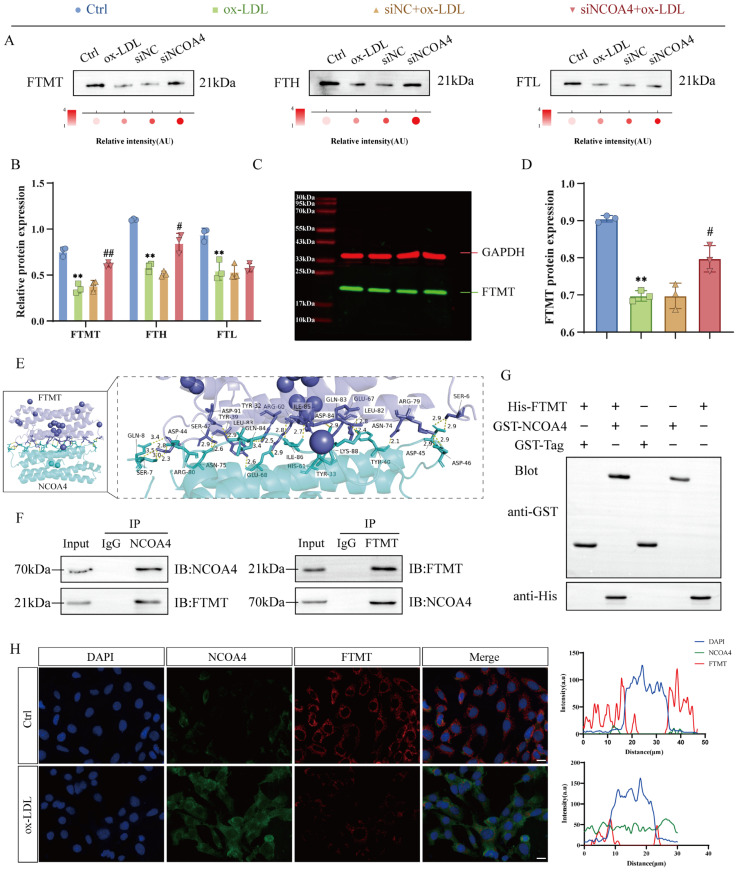
NCOA4 facilitates FTMT degradation via direct interaction. (**A**,**B**) Western blot to detect the expression of FTMT, FTH, and FTL in the VSMCs (*n* = 3). (**C**,**D**) Near-infrared dual-color laser imaging system to detect the expression of FTMT (*n* = 3). (**E**) Molecular docking plot of NCOA4 with FTMT protein. (**F**) Endogenous immunoprecipitation (IP) analysis of NCOA4 and FTMT in VSMCs. (**G**) The direct interaction between GST-NCOA4 and His-FTMT was confirmed through GST pull-down assays. (**H**) Representative immunofluorescence images illustrating the presence of NCOA4 and FTMT in VSMCs (scale bar: 100 µm, *n* = 3). ** *p* < 0.01 vs. control group, ^#^
*p* < 0.05, ^##^
*p* < 0.01 vs. ox-LDL group.

**Figure 4 nutrients-17-03713-f004:**
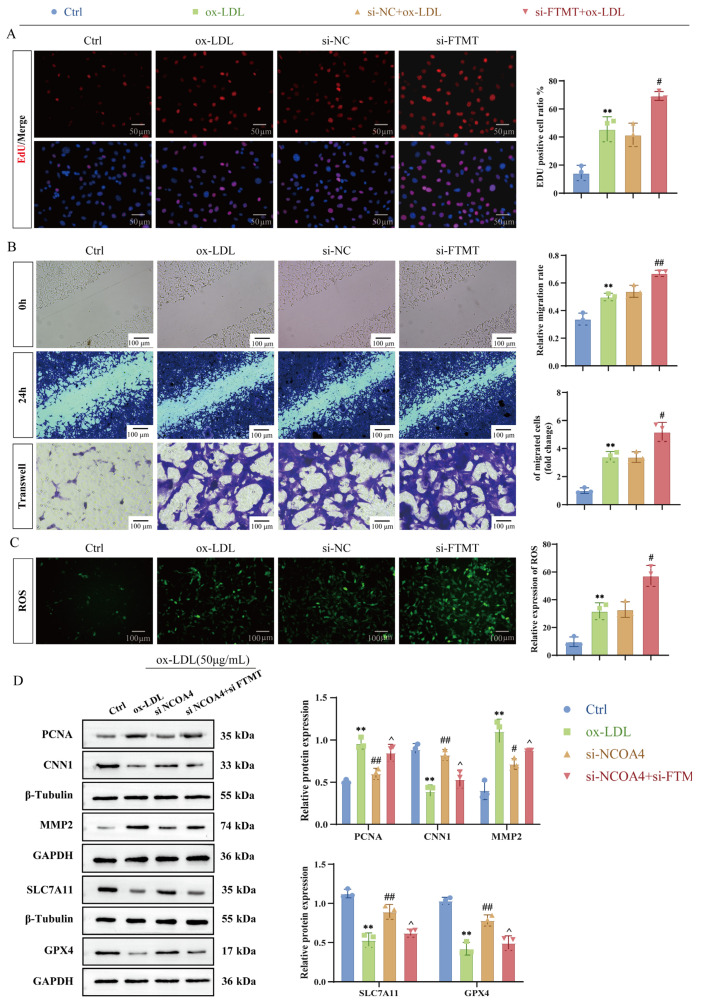
VSMCs-specific FTMT gene knockout enhances VSMCs proliferation and migration. (**A**) Proliferation analysis of mouse VSMCs was performed using fluorescence EdU detection (red) and DAPI staining (blue) (scale bar: 50 µm, *n* = 3). (**B**) The migratory capacity of mouse VSMCs was evaluated through transwell assays and scratch wound healing experiments, measuring the width of the scratch gap and the migration distance of VSMCs at both 0 h and 24 h (scale bar: 100 µm, *n* = 3). (**C**) Intracellular levels of ROS were assessed utilizing immunofluorescence techniques (scale bar: 100 µm, *n* = 3). (**D**) Expression levels of PCNA, MMP2, CNN1, and ferroptosis-related proteins in VSMCs were analyzed via Western blotting. ** *p* < 0.01 vs. control group, ^#^
*p* < 0.05, ^##^
*p* < 0.01 vs. ox-LDL group, ^ *p* < 0.05 vs. Si-NCOA4+ox-LDL group.

**Figure 5 nutrients-17-03713-f005:**
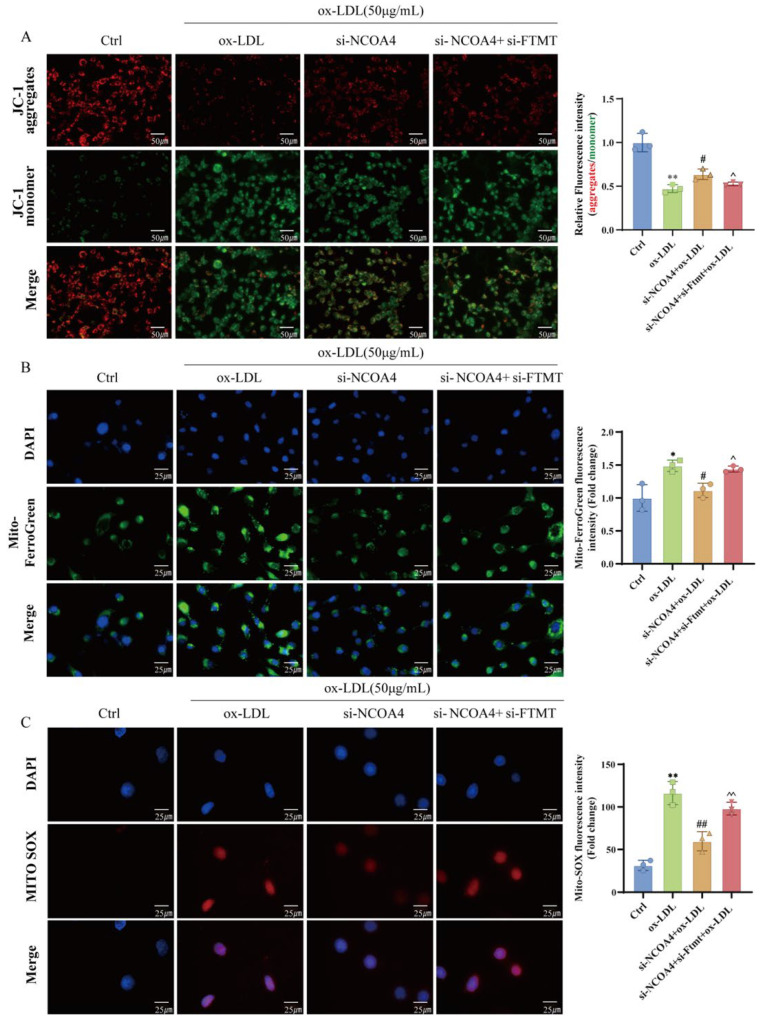
NCOA4 induces mitochondrial damage in VSMCs via FTMT suppression and iron homeostasis dysregulation. (**A**) JC-1 staining for mitochondrial membrane potential measurement and quantification of membrane potential by measuring the ratio of red and green fluorescence intensity graph (scale bar: 50 µm, *n* = 3). (**B**) Mitochondrial Fe^2+^ was assessed by Mito-FerroGreen probe fluorescence intensity; DAPI was used to stain nuclei (scale bar: 25 µm, *n* = 3). (**C**) Mitochondrial ROS was assessed by MitoSOX probe fluorescence intensity (scale bar: 25 µm, *n* = 3). * *p* < 0.05, ** *p* < 0.01 vs. control group, ^#^
*p* < 0.05, ^##^
*p* < 0.01 vs. ox-LDL group, ^ *p* < 0.05, ^^ *p* < 0.01 vs. si-NCOA4+ox-LDL group.

**Figure 6 nutrients-17-03713-f006:**
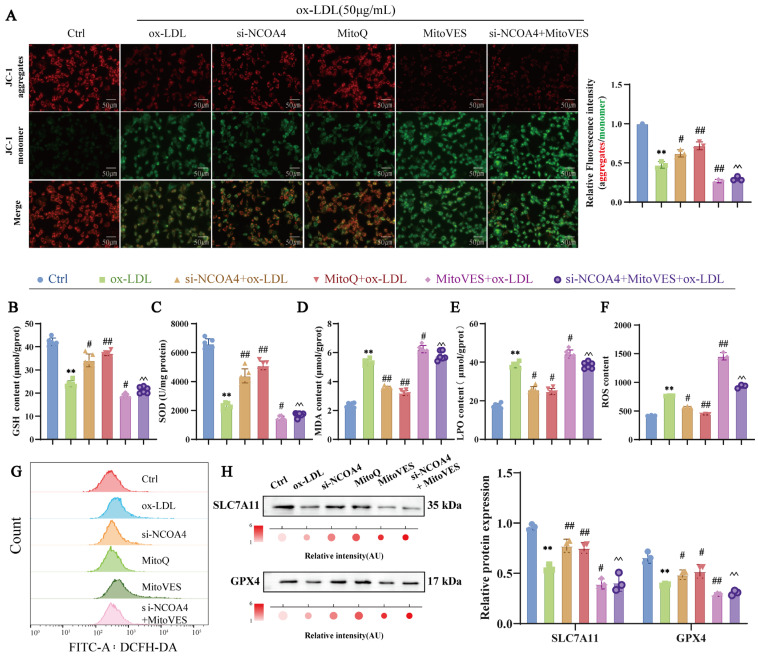
Silencing NCOA4 inhibits ferroptosis in VSMCs by improving mitochondrial damage. (**A**) JC-1 staining for mitochondrial membrane potential measurement and quantification of membrane potential by measuring the ratio of the red and green fluorescence intensity graph (scale bar: 50 µm, *n* = 3). (**B**–**E**) Biochemical kit for GSH, SOD MDA, and LPO levels (*n* = 6). (**F**,**G**) Flow cytometry for ROS levels (*n* = 3). (**H**) Western blot to detect SLC7A11 and GPX4 proteins (*n* = 3). ** *p* < 0.01 vs. control group, ^#^
*p* < 0.05, ^##^
*p* < 0.01 vs. ox-LDL group, ^^ *p* < 0.01 vs. si-NCOA4 + ox-LDL group.

**Figure 7 nutrients-17-03713-f007:**
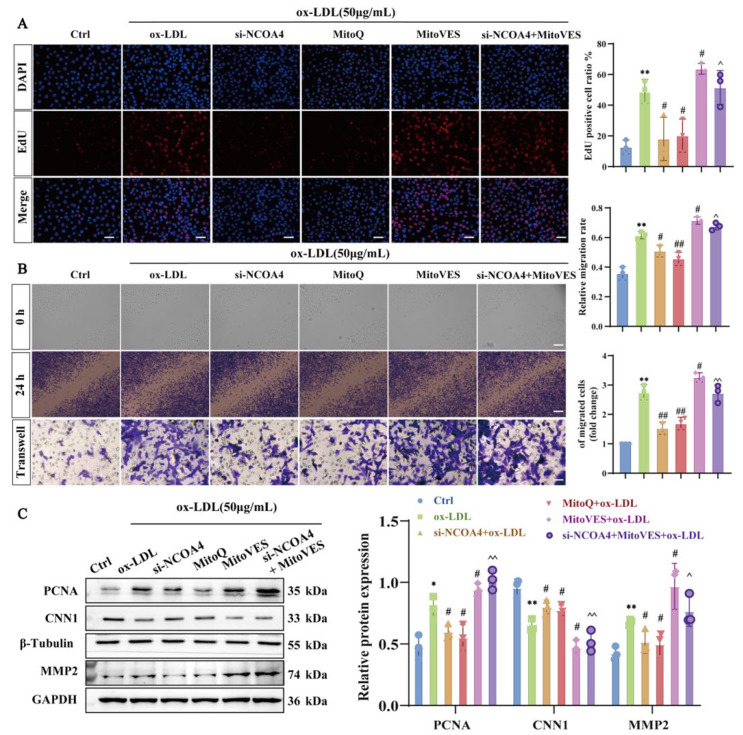
Silencing NCOA4 inhibits ferroptosis in VSMCs by improving mitochondrial damage. (**A**) Proliferation analysis in mouse VSMCs using fluorescent EdU assay (scale bar: 50 µm, *n* = 3). (**B**) Evaluation of VSMCs migration using transwell assay and scratch assay in mouse VSMCs (scale bar: 100 µm, *n* = 3). (**C**) Western blot to detect PCNA, MMP2, SLC7A11 and GPX4 proteins (*n* = 3). * *p* < 0.05, ** *p* < 0.01 vs. Ctrl group, ^#^
*p* < 0.05, ^##^
*p* < 0.01 vs. ox-LDL group, ^ *p* < 0.05, ^^ *p* < 0.01 vs. Si-NCOA4+ox-LDL group.

**Figure 8 nutrients-17-03713-f008:**
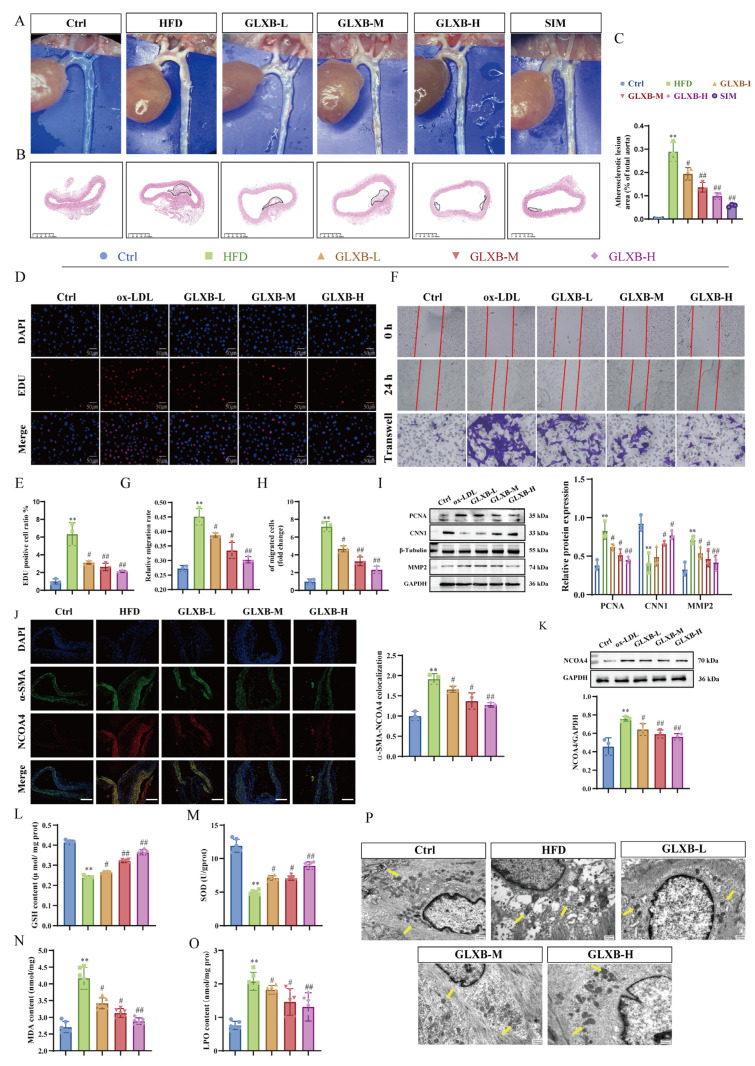
GLXB attenuates atherosclerotic plaque formation via NCOA4-dependent suppression of ferroptosis in VSMCs. (**A**) Representative en face aortic images illustrating lipid deposition across the experimental groups. (**B**,**C**) HE-stained cross-sections of aortic plaques (scale bar: 200 µm, *n* = 3; (**C**): quantitative analysis). (**D**,**E**) EdU incorporation assay evaluating VSMCs proliferation (scale bar: 50 µm, *n* = 3; (**E**): quantitative analysis). (**F**) VSMCs migration assessed through transwell assays and scratch wound healing assays (scale bar: 100 µm, *n* = 3; (**G**,**H**) quantitative analysis). (**I**) Western blot analysis was used to detect the protein expression levels of PCNA, MMP2, and CNN1 in VSMCs (*n* = 3). (**J**) Immunofluorescence colocalization of α-SMA (green) and NCOA4 (red) in aortic VSMCs (*n* = 3, scale bar: 100 μm). (**K**) Western blot analysis was used to detect the protein levels of NCOA4 (*n* = 3). (**L**–**O**) Measurement of aortic lipid peroxidation biomarkers utilizing biochemical kits: MDA (**L**), LPO (**M**), GSH (**N**), and SOD (**O**) activity levels were assessed (*n* = 6). (**P**) TEM images demonstrating mitochondrial ultrastructural changes in aortic VSMCs (yellow arrows indicate mitochondria; scale bar: 500 nm, *n* = 3). ** *p* < 0.01 vs. Ctrl group, ^#^ *p* < 0.05, ^##^ *p* < 0.01 vs. ox-LDL group.

**Figure 9 nutrients-17-03713-f009:**
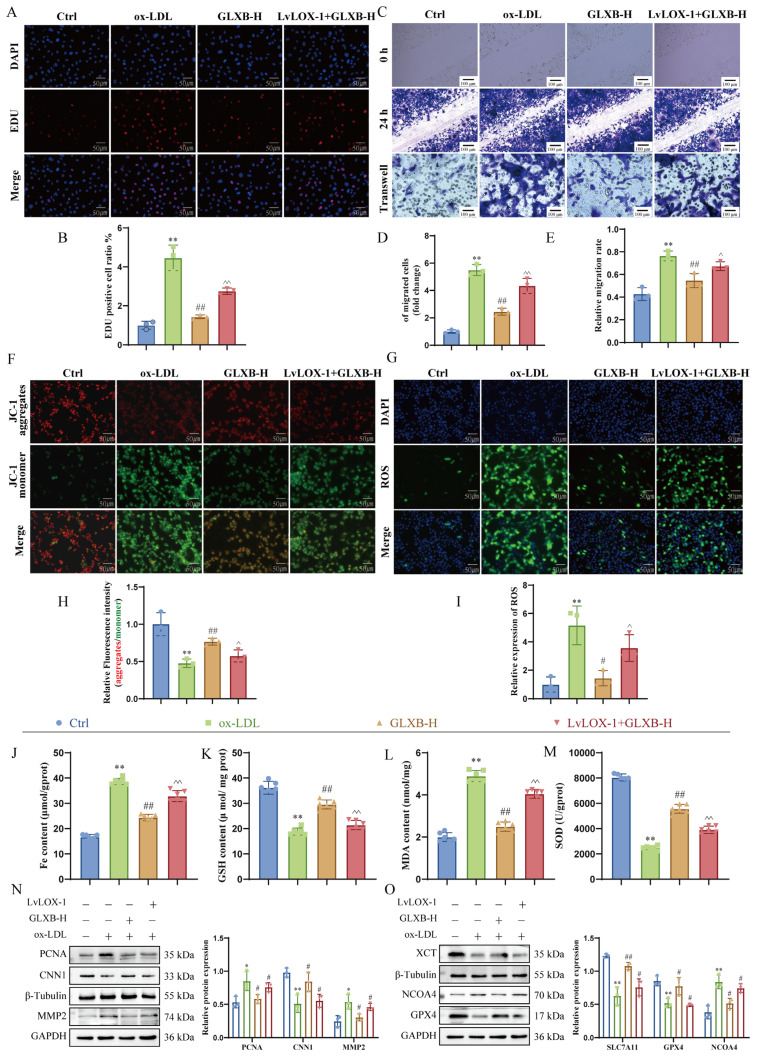
GLXB-containing serum attenuates VSMCs proliferation and migration by targeting LOX-1 to suppress NCOA4-dependent ferroptosis. (**A**,**B**) Proliferation analysis in mouse VSMCs using fluorescent EdU assay (scale bar: 50 µm, *n* = 3; (**B**): quantitative analysis). (**C**–**E**) Evaluation of VSMC migration using transwell assay and scratch assay in mouse VSMCs (scale bar: 100 µm, *n* = 3; (**D**,**E**): quantitative analysis). (**F**) JC-1 staining for mitochondrial membrane potential measurement and quantification of membrane potential by measuring the ratio of red and green fluorescence intensity graph (scale bar: 50 µm, *n* = 3; (**H**): quantitative analysis). (**G**) Intracellular levels of ROS were assessed utilizing immunofluorescence techniques (scale bar: 50 µm, *n* = 3; (**I**): quantitative analysis). (**J**–**M**) Measurement of aortic lipid peroxidation biomarkers utilizing biochemical kits: Fe (**J**), GSH (**K**), MDA (**L**), and SOD (**M**) activity levels were assessed (*n* = 6). (**N**,**O**) Western blot to detect PCNA, MMP2, CNN1, SLC7A11, GPX4 and NCOA4 proteins (*n* = 3). * *p* < 0.05, ** *p* < 0.01 vs. Ctrl group, ^#^
*p* < 0.05, ^##^
*p* < 0.01 vs. ox-LDL group, ^ *p* < 0.05, ^^ *p* < 0.01 vs. GLXB-H group.

## Data Availability

The original contributions presented in this study are included in the article/Appendix A. Further inquiries can be directed to the corresponding author.

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
