# Peer review of "Ferritin Mitochondrial (FTMT)-Driven Mitochondrial Ferroptosis in Vascular Smooth Muscle Cells: A Role of NCOA4 in Atherosclerosis Pathogenesis and Modulation by Gualou–Xiebai"

_nutrients, 2025, doi:10.3390/nu17233713_

Round 1

Reviewer 1 Report

Comments and Suggestions for Authors

This is a very interesting work presented by the authors.

I kindly invite you to revisit the following points:

(1) Lines 114-115: "Mice were randomly assigned to groups and administered a tail vein injection of the vehicle control (PBS with 0.1% BSA) once every 4 weeks for a total of 4 weeks". Please revisit the wording here, since, as it is, the reader may be confused regarding the duration of the study. Perhaps "once every 4 weeks for a total of 16 weeks" is what the authors had in mind.

(2) In the methodology section, please provide appropriate references to the study protocols for 2.3, 2.5, 2.7 to 2.10, and 2.12-2.15 in order to support the selection of the time frame for preparation and implementation, reagent amounts etc.. If the study protocols use in-house methods, please consider mentioning this in the manuscript and providing a detailed description.

(3) Results: Kindly consider including (in supplementary) the numbers describing the following graphs: Figure 1-B, 1-E to I, Figure 2-A to D, Figure 3-B and D, (Also consider providing image 3-E in a readable size), Figure 4-A to D, Figure 5-A to C, Figure 6-A to F and 6-H, Figure 7-A to C. The availability of these data ensures that the results are available for the reader and given the small size of the graphs, this allows a clear understanding of the intervention's impact. Also, in terms of future research, a detailed presentation of the outcomes allow the reproduction of similar experiments that will help moving forward in this research field. Please consider giving this proper attention. It is clear that the research is detailed and the data would be of great value to build on.

(4) In the methodology section the following groups are described: (a) high fat diet (HFD); (b) HFD + LvNCOA4; (c) HFD+ LvNC; (d) HFD + shNCOA4; (e) HFD + shNC in addition to the Control group (normal diet). Kindly consider elaborating if the outcomes of groups (c) and (e) were evaluated.

(5) Erastin treatment is reported in the outcomes, however the treatment protocol is not available in the methodology. Kindly consider elaborating on this in the approprate segements of the methodology in order to help the reader have a clear understading of the outcomes and follow the manuscript's narrative. (same for Fer-1 and MitoQ).

(6) Please revisit the spelling of Erastin in Figure 2

Author Response

Dear Editors and Reviewers:

Thank you for your letter and comments concerning our manuscript entitled Ferritin mitochondrial (FTMT)-driven mitochondrial ferroptosis in vascular smooth muscle cells: a role of NCOA4 in atherosclerosis pathogenesis and modulation by Gualou-Xiebai (Manuscript ID: Nutrients -3969812). Those comments are all valuable and helpful for improving our paper. According to the reviewer’ comments, we have made extensive modifications to our manuscript in this revised version and the revised portions are marked in red in the paper. All changes made are marked in red in the paper. And point-by-point responses to the comments are provided below. We hope that the revision will meet with approval, and your favorable consideration of our manuscript is greatly appreciated.

Best wishes.

Yours, sincerely

Hongfei Wu

Responses to the Reviewers Comments

Dear Editors and Reviewers:

Thank you for your letter and comments concerning our manuscript entitled Ferritin mitochondrial (FTMT)-driven mitochondrial ferroptosis in vascular smooth muscle cells: a role of NCOA4 in atherosclerosis pathogenesis and modulation by Gualou-Xiebai (Manuscript ID: Nutrients -3969812). Those comments are all valuable and helpful for improving our paper. According to the reviewer’ comments, we have made extensive modifications to our manuscript in this revised version and the revised portions are marked in red in the paper. All changes made are marked in red in the paper. And point-by-point responses to the comments are provided below. We hope that the revision will meet with approval, and your favorable consideration of our manuscript is greatly appreciated.

Best wishes.

Yours, sincerely

Hongfei Wu

Responses to the Reviewers Comments

We deeply appreciate the comments from Reviewer 1# on our manuscript

Nutrients -3969812. The following are our responses.

Reviewer #1 (Remarks to the Author):

General Comments: This is a very interesting work presented by the authors. I kindly invite you to revisit the following points:

Major Comment 1: Lines 114-115: "Mice were randomly assigned to groups and administered a tail vein injection of the vehicle control (PBS with 0.1% BSA) once every 4 weeks for a total of 4 weeks". Please revisit the wording here, since, as it is, the reader may be confused regarding the duration of the study. Perhaps "once every 4 weeks for a total of 16 weeks" is what the authors had in mind.

Response: We thank the reviewer for pointing out the confusion in our wording. The original description was inaccurate. To clarify, the mice did not receive injections every 4 weeks for the entire study duration. Instead, they received two tail vein injections specifically at week 4 and week 8, and the observation period continued until week 16. This booster injection strategy was employed to ensure robust and sustained transgene expression.

The manuscript has been revised for precision. Lines 114-115 now read: Mice were randomly assigned to groups and administered tail vein injections of the vehicle control (PBS with 0.1% BSA) at week 4 and week 8, and the observation period continued until week 16.

Major Comment 2: In the methodology section, please provide appropriate references to the study protocols for 2.3, 2.5, 2.7 to 2.10, and 2.12-2.15 in order to support the selection of the time frame for preparation and implementation, reagent amounts etc.. If the study protocols use in-house methods, please consider mentioning this in the manuscript and providing a detailed description.

Response: We sincerely thank the reviewer for this critical suggestion. Providing authoritative references for experimental protocols and clarifying in-house optimized methods are essential for ensuring the reproducibility and rigor of the methodology. As requested, we have supplemented appropriate references for Sections 2.3, 2.5, 2.7–2.10, and 2.12–2.15, and detailed the in-house optimized parameters.

Major Comment 3: Results: Kindly consider including (in supplementary) the numbers describing the following graphs: Figure 1-B, 1-E to I, Figure 2-A to D, Figure 3-B and D, (Also consider providing image 3-E in a readable size), Figure 4-A to D, Figure 5-A to C, Figure 6-A to F and 6-H, Figure 7-A to C. The availability of these data ensures that the results are available for the reader and given the small size of the graphs, this allows a clear understanding of the intervention's impact. Also, in terms of future research, a detailed presentation of the outcomes allow the reproduction of similar experiments that will help moving forward in this research field. Please consider giving this proper attention. It is clear that the research is detailed and the data would be of great value to build on.

Response: We sincerely appreciate the reviewer’s constructive and critical comment, which aptly highlights the significance of data accessibility and transparency to the scientific community. We fully concur with this perspective, greatly value this feedback, and have proactively addressed it: detailed numerical data corresponding to all requested figures have now been compiled and included in the Supplementary Information (SI) as a structured table (Table S1).

Major Comment 4: In the methodology section the following groups are described: (a) high fat diet (HFD); (b) HFD + LvNCOA4; (c) HFD+ LvNC; (d) HFD + shNCOA4; (e) HFD + shNC in addition to the Control group (normal diet). Kindly consider elaborating if the outcomes of groups (c) and (e) were evaluated.

Response: We thank the reviewer for raising this important point regarding the experimental design. We confirm that all groups described in the Methods, including the negative control groups (c) HFD + LvNC (empty vector overexpression control) and (e) HFD + shNC (scrambled shRNA control), were fully evaluated and analyzed in our study.

These control groups are critical to account for any potential non-specific effects associated with the lentiviral vector backbone and the viral infection procedure itself. The comparisons between HFD + LvNC vs. HFD + LvNCOA4 and HFD + shNC vs. HFD + shNCOA4 allow us to confidently attribute the observed phenotypic differences—such as changes in plaque area and VSMC phenotypes—to the specific manipulation of NCOA4 expression, rather than to off-target effects.

For the sake of clarity and focus in the main figures, the primary validation data for these negative controls are presented in the Supplementary Information. Specifically, Supplementary Figure S1A demonstrates the efficacy and specificity of our manipulations by comparing NCOA4 protein expression levels in these control groups versus their corresponding experimental groups.

Supplementary Fig. 1. NCOA4-mediated ferroptosis inhibits the proliferation and migration of VSMCs. (A) The validation of NCOA4 lentiviral transfection efficiency (n=3). (B) Aortic imaging showing aortic lipid deposition in each group. (C-D) HE staining of aortic in each group (scale bar: 200µm, D: quantitative analysis). (E) Quantitative analysis of the co-localization of α-SMA (red)and GPX4 (green) in atherosclerotic lesions. *P < 0.05, **P < 0.01 vs. control group; #P < 0.05, ##P < 0.01 vs. HFD group.

Major Comment 5: Erastin treatment is reported in the outcomes, however the treatment protocol is not available in the methodology. Kindly consider elaborating on this in the approprate segements of the methodology in order to help the reader have a clear understading of the outcomes and follow the manuscript's narrative. (same for Fer-1 and MitoQ).

Response: We thank the reviewer for this valuable suggestion. We have now comprehensively elaborated on the treatment protocols for Erastin, Fer-1, MitoQ, and other compounds in the “2. Materials and Methods section section (subsection 2.5. Cell culture and Groupings).

2.5. Cell culture and Groupings

Mouse vascular smooth muscle cells were maintained in high-glucose DMEM supplemented with 10% FBS and 1% penicillin-streptomycin at 37°C with 5% CO₂. After serum starvation, VSMCs were divided into 12 groups. The Control group was cultured in 10% DMEM, while the ox-LDL group was stimulated with 50 μg/mL ox-LDL for 24 h. The ox-LDL + Fer-1 group received 4 h pre-treatment with 1 μmol/L Fer-1 followed by 24 h co-incubation with ox-LDL. The ox-LDL + Erastin group was co-treated with 10 μmol/L Erastin and ox-LDL for 24 h. siRNA transfection groups (siNC, siNCOA4, siFTMT, and co-transfected siNCOA4+siFTMT) were stimulated with ox-LDL 48 h post-transfection for 24 h. The ox-LDL + siNCOA4 + Erastin group was co-treated with Erastin and ox-LDL post-transfection. The ox-LDL + MitoQ group was pre-treated with 100 nmol/L MitoQ for 2 h before ox-LDL co-incubation. The MitoVES-containing groups (ox-LDL + MitoVES and ox-LDL + siNCOA4 + MitoVES) were treated with 5 μmol/L MitoVES and ox-LDL for 24 h. Vehicle controls were included. The manuscript has been updated with these details.

Major Comment 6:Please revisit the spelling of Erastin in Figure 2

We are grateful to the reviewer for their thorough reading of our manuscript and for identifying this typographical error. We have carefully corrected the spelling of "Erastin" throughout the manuscript, including in the legend of Figure 2 and the main text of Section 3.2. The revised version of the manuscript, which incorporates this correction, has been resubmitted.

Figure 2. NCOA4 knockdown inhibits ferroptosis and reduces ox-LDL-induced VSMCs proliferation and migration. (A) Proliferation analysis in mouse VSMCs using fluorescent EdU assay (red), DAPI was used to stain nuclei (blue) (n =3). (B) Evaluation of VSMCs migration using transwell assay and wound healing assay (scratch assay) in mouse VSMCs. The gap created by scratch at two time-points (0 h and 24 h) and the migration of VSMCs was calculated (n=3). (C-D) Western blotting to detect the expression of PCNA, MMP2, CNN1 and ferroptosis-related protein in VSMCs. (E) Immunofluorescence to detect intracellular ROS level (n=3). (F) Flow cytometry for ROS levels (n = 3). *P < 0.05, **P < 0.01 vs. control group, #P < 0.05, ##P < 0.01 vs. ox-LDL group, ^P < 0.05, ^^P < 0.01 vs. si-NCOA4+ox-LDL group.

Reviewer 2 Report

Comments and Suggestions for Authors

The manuscript entited FTMT-dependent mitochondria-mediated ferroptosis in vascular smooth muscle cells: NCOA4 as a critical regulator underlying atherosclerosis pathogenesis is an original paper. The authors assessed whether ferroptosis in vascular smooth muscle cells (VSMCs) influences their proliferation and migration in atherosclerosis, and to elucidate the underlying mechanisms involving nuclear receptor coactivator 4 (NCOA4). They used an ApoE-/- mouse model fed a high-fat diet to induce atherosclerosis and they studied the progression of atherosclerotic and markers of mitochondrial ferroptosis by histological and biochemical analyses. The authors concluded that NCOA4-mediated ferroptosis occurs in both ApoE-/- mice and VSMCs.

The fact that this study proved that NCOA4 plays a critical role in promoting atherosclerosis progression is consistent with previous research reports. Also, they underlined the essential role of NCOA4-FTMT interaction in coordinating phenotypic transitions in vascular smooth muscle cells. However, the authors use frequently some expressions like : ‘’These findings suggest that..’’ or ‘’ This study elucidates potential mechanisms….’’ which means that there are only some associations not cauzal relationship. Therefore, is not appropriate saying that: ‘’Thus, we have established that silencing NCOA4 can mitigate ferroptosis by upregulating FTMT expression…’’

The manuscript is well written. Materials and methods are very well detailed. Results are well presented. This study is important because open others possibility to research.

What are the study limitations ? Please specify.

Please explain the acronym NCOA4 in abstract.

Author Response

Dear Editors and Reviewers:

Thank you for your letter and comments concerning our manuscript entitled Ferritin mitochondrial (FTMT)-driven mitochondrial ferroptosis in vascular smooth muscle cells: a role of NCOA4 in atherosclerosis pathogenesis and modulation by Gualou-Xiebai (Manuscript ID: Nutrients -3969812). Those comments are all valuable and helpful for improving our paper. According to the reviewer’ comments, we have made extensive modifications to our manuscript in this revised version and the revised portions are marked in red in the paper. All changes made are marked in red in the paper. And point-by-point responses to the comments are provided below. We hope that the revision will meet with approval, and your favorable consideration of our manuscript is greatly appreciated.

Best wishes.

Yours, sincerely

Hongfei Wu

We deeply appreciate the comments from Reviewer 2# on our manuscript

Nutrients -3969812. The following are our responses.

Reviewer #2 (Remarks to the Author):

General Comments: The manuscript entited FTMT-dependent mitochondria-mediated ferroptosis in vascular smooth muscle cells: NCOA4 as a critical regulator underlying atherosclerosis pathogenesis is an original paper. The authors assessed whether ferroptosis in vascular smooth muscle cells (VSMCs) influences their proliferation and migration in atherosclerosis, and to elucidate the underlying mechanisms involving nuclear receptor coactivator 4 (NCOA4). They used an ApoE-/- mouse model fed a high-fat diet to induce atherosclerosis and they studied the progression of atherosclerotic and markers of mitochondrial ferroptosis by histological and biochemical analyses. The authors concluded that NCOA4-mediated ferroptosis occurs in both ApoE-/- mice and VSMCs.

The fact that this study proved that NCOA4 plays a critical role in promoting atherosclerosis progression is consistent with previous research reports. Also, they underlined the essential role of NCOA4-FTMT interaction in coordinating phenotypic transitions in vascular smooth muscle cells. However, the authors use frequently some expressions like: ‘’These findings suggest that..’’ or ‘’ This study elucidates potential mechanisms….’’ which means that there are only some associations not cauzal relationship. Therefore, is not appropriate saying that: ‘’Thus, we have established that silencing NCOA4 can mitigate ferroptosis by upregulating FTMT expression…’’

The manuscript is well written. Materials and methods are very well detailed. Results are well presented. This study is important because open others possibility to research.

Response: We thank the reviewer for the positive comments. 

Major Comment 1: What are the study limitations? Please specify.

Response: We thank the reviewer for this valuable suggestion. A transparent discussion of study limitations is critical for contextualizing findings and guiding future research directions. Inspired by your valuable insights, I have added this perspective into the DISCUSSION section (lines 704-711 on pages 25).

We acknowledge several limitations of the present work that warrant attention. First, while our data demonstrate a functional and physical interaction between NCOA4 and FTMT, the exact molecular pathway mediating FTMT degradation remains incompletely defined. We provide evidence for NCOA4-dependent FTMT downregulation, but future studies are needed to clarify whether this process relies on the canonical autophagy-lysosome pathway (e.g., mitophagy, given FTMT’s mitochondrial localization) or alternative protease systems (e.g., proteasomal degradation). Definitive validation—such as using pathway-specific inhibitors (e.g., bafilomycin A1 for autophagy, MG132 for proteasomes) or genetic ablation of key pathway components—will be required to confirm the mechanism.

These limitations do not undermine the core findings of our study but highlight important avenues for future research. We believe addressing these questions will deepen our understanding of the NCOA4-FTMT axis in AS and facilitate the development of more targeted therapeutic strategies. We appreciate the reviewer’s guidance in strengthening the rigor and completeness of our discussion.

Major Comment 2: Please explain the acronym NCOA4 in abstract.

Response: We thank the reviewer for this suggestion. In response, We have defined the acronym NCOA4 as "Nuclear Receptor Coactivator 4" in the Revised Manuscript (Lines 18, Page 1 )

Reviewer 3 Report

Comments and Suggestions for Authors

In the current study the authors investigated whether ferroptosis in vascular smooth muscle cells (VSMCs) influences their proliferation and migration, and elucidated the underlying  mechanisms involving nuclear receptor coactivator (NCOA4) during atherosclerosis progression. They concluded that NCOA4 exacerbates atherosclerotic progression by binding to and repressing FTMT, thereby disrupting mitochondrial iron homeostasis and promoting ferroptosis in VSMCs. As a conclusion, the study established a novel mechanistic link between NCOA4, ferroptosis, and VSMCs dysfunction in atherosclerosis.

Some suggestions:

  1. The use of the abbreviation in the title is not recommended.

2.Lines 63-66, you wrote: “Notably, ferritin mitochondrial (FTMT), a key regulatory factor in maintaining mitochondrial iron homeostasis, helps preserve the redox balance within mitochondria by chelating free  iron ions”. Add please some more details.

  1. Point2.2. Animals and Grouping:

- you had Sixty male ApoE-/- mice and you wrote about 5 groups = 50 mice. The 6th group represents the control = 10 mice? Please clarify.

- which is te provenance of LvNCOA4, shNCOA4 and shNC? Please add.

-you must specify how the mice were kept during the experiment.

-Please add under which conditions the serum and aorta where stored until they were analyzed.

  1. Line 133-134, you wrote: “VSMCs were treated with ox-LDL (50μg/mL) to induce VSMCs proliferation and migration in vitro”. Give please some details.
  2. Line 147-148: Please specify the type of stereomicroscope.
  3. Point 2.7. Hematoxylin-eosin (HE) staining:

-lines 151-152: what do you mean by “ethanol of different concentration  gradients”. Please clarify.

-line 156 – add please the type of microscope.

7.line 163 / line 172  -add please the type of laser confocal microscope / the type of fluorescence microscope

  1. Point 2.10. Migration assay:

-lines 176-77, you wrote “Subsequently, different treatments were applied”. Please clarify

- with what were the photos taken?

  1. Lines 186 – please add how the Total protein was extracted from aortic tissue and VSMCs.

10.line 197 – what skim milk? Please clarify.

11.Point 2.14 Biochemical analysis:

-lines 211-212, you wrote “different treatments were carried out according to the        requirements of the experimental group”. Please mention which are the different treatments.

-which biochemical kits were used to determine GSH/SOD/MDA/Fe content? Please add.

12.Lines 100-102, you wrote “Gualou and Xiebai were purchased from Beijing Tong ren tang Co., Ltd. (Beijing, China; batch number: 211001, 211001”. For what did you need these? Please clarify. 13.One of the added key words is personalized nutrition. In the article you didn’t write anything about this aspect. How do you justify to add it as a key word? Please clarify. 

At discussions you wrote only: “Therefore, regulating the proliferation and migration of VSMCs is essential for enhancing therapeutic strategies aimed at treating AS”.

 14. Add please the limitations of the study.   

Author Response

Dear Editors and Reviewers:

Thank you for your letter and comments concerning our manuscript entitled Ferritin mitochondrial (FTMT)-driven mitochondrial ferroptosis in vascular smooth muscle cells: a role of NCOA4 in atherosclerosis pathogenesis and modulation by Gualou-Xiebai (Manuscript ID: Nutrients -3969812). Those comments are all valuable and helpful for improving our paper. According to the reviewer’ comments, we have made extensive modifications to our manuscript in this revised version and the revised portions are marked in red in the paper. All changes made are marked in red in the paper. And point-by-point responses to the comments are provided below. We hope that the revision will meet with approval, and your favorable consideration of our manuscript is greatly appreciated.

Best wishes.

Yours, sincerely

Hongfei Wu

We deeply appreciate the comments from Reviewer 3# on our manuscript

Nutrients -3969812. The following are our responses.

Reviewer #3 (Remarks to the Author):

General Comments: In the current study the authors investigated whether ferroptosis in vascular smooth muscle cells (VSMCs) influences their proliferation and migration, and elucidated the underlying mechanisms involving nuclear receptor coactivator (NCOA4) during atherosclerosis progression. They concluded that NCOA4 exacerbates atherosclerotic progression by binding to and repressing FTMT, thereby disrupting mitochondrial iron homeostasis and promoting ferroptosis in VSMCs. As a conclusion, the study established a novel mechanistic link between NCOA4, ferroptosis, and VSMCs dysfunction in atherosclerosis.

Response: We thank the reviewer for the positive comments.

Some suggestions:

Major Comment 1: The use of the abbreviation in the title is not recommended.

Response: We thank the reviewer for raising this important point regarding abbreviation use in the title. We agree with the general principle of minimizing non-standard abbreviations for clarity. In this case, we chose to retain "NCOA4" as it is the established and universally recognized gene symbol in the field, particularly in the context of ferritinophagy and ferroptosis. Replacing it with its full name, "Nuclear Receptor Coactivator 4," would make the title noticeably longer and less concise. Conversely, we have spelled out the less common abbreviation "FTMT" (Ferritin Mitochondrial) in full to introduce this key new component of our study. We believe the current title, "Ferritin mitochondrial (FTMT)-driven mitochondrial ferroptosis in vascular smooth muscle cells: a role for NCOA4 in atherosclerosis pathogenesis," effectively balances precision with readability. As standard practice, both "NCOA4" and "FTMT" are defined upon their first appearance in the abstract and main text.

Major Comment 2: Lines 63-66, you wrote: “Notably, ferritin mitochondrial (FTMT), a key regulatory factor in maintaining mitochondrial iron homeostasis, helps preserve the redox balance within mitochondria by chelating free iron ions”. Add please some more details.

Response: We thank the reviewer for this constructive suggestion. We have now expanded the description of mitochondrial ferritin (FTMT) in the revised manuscript to provide deeper mechanistic insight into its role in ferroptosis regulation. The red part is the detail we added in this revised manuscript): (lines 64-73, page 2).

  1. Introduction

Ferroptosis execution is mechanistically governed by the integrated dysregulation of redox homeostasis, iron metabolism, phospholipid peroxidation, and mitochondrial bioenergetics, pathologically hallmarked by the lethal accumulation of reactive oxygen species (ROS) and membrane-localized toxic lipid peroxidation derivatives[8,9]. Mitochondria serve as central hubs for cellular iron and oxidative metabolism, playing a crucial role in ROS production; their functional state is intricately linked to ferroptosis[10,11]. While ROS can function as signaling molecules, elevated levels are cytotoxic and may induce oxidative cell death[12]. Evidence indicates that alterations in mitochondrial morphology—such as mitochondrial fission and cristae expansion—are closely associated with ferroptosis, with several effective inhibitors targeting mitochondria directly[10,13]. Notably, mitochondrial ferritin (FTMT), a key guardian of mitochondrial iron homeostasis, functions by chelating free iron ions to preserve the redox balance[14]. This nuclear-encoded protein is specifically targeted to the mitochondrial matrix, where it forms a ferroxidase-active cage structure to store iron safely. In contrast to cytosolic ferritin, FTMT exhibits low basal expression that is upregulated under stress, pinpointing its role as a dedicated mitochondrial iron buffer[15,16]. Its primary protective mechanism lies in sequestering the reactive labile iron pool, which directly limits the substrate available for the Fenton reaction, thus preventing the generation of deleterious reactive oxygen species and preserving mitochondrial function[17]. However, our current understanding of the molecular regulatory mechanisms involving FTMT in ferroptosis remains limited.

References

[14]. Fuhrmann, D.C.; Mondorf, A.; Beifuß, J.; Jung, M.; Brüne, B. Hypoxia inhibits ferritinophagy, increases mitochondrial ferritin, and protects from ferroptosis. Redox biology 2020, 36, 101670, doi:10.1016/j.redox.2020.101670.

[15]. Bradley, J.M.; Bugg, Z.; Pullin, J.; Moore, G.R.; Svistunenko, D.A.; Le Brun, N.E. Human mitochondrial ferritin exhibits highly unusual iron-O(2) chemistry distinct from that of cytosolic ferritins. Nature communications 2025, 16, 4695, doi:10.1038/s41467-025-59463-1.

[16]. Wang, P.; Cui, Y.; Ren, Q.; Yan, B.; Zhao, Y.; Yu, P.; Gao, G.; Shi, H.; Chang, S.; Chang, Y.Z. Mitochondrial ferritin attenuates cerebral ischaemia/reperfusion injury by inhibiting ferroptosis. Cell death & disease 2021, 12, 447, doi:10.1038/s41419-021-03725-5.

[17]. Wang, P.; Cui, Y.; Liu, Y.; Li, Z.; Bai, H.; Zhao, Y.; Chang, Y.Z. Mitochondrial ferritin alleviates apoptosis by enhancing mitochondrial bioenergetics and stimulating glucose metabolism in cerebral ischemia reperfusion. Redox biology 2022, 57, 102475, doi:10.1016/j.redox.2022.102475.

Major Comment 3: Point2.2. Animals and Grouping:

- you had Sixty male ApoE-/- mice and you wrote about 5 groups = 50 mice. The 6th group represents the control = 10 mice? Please clarify.

Response: Thank you for pointing out the mistakes. The study indeed used a total of 60 mice, comprising 50 ApoE-/- mice distributed across five experimental groups, and 10 wild-type C57BL/6J mice serving as the genetic control group. We revised this part in the “2.2. Animals (Lines 114, page 3).

2.2. Animals

Fifty male ApoE-/- mice and ten male wild-type C57BL/6J mice (6-8 weeks old, 20-22 g) were obtained from Jiangsu Hua Chuang Xin Nuo Pharmaceutical Technology Co., Ltd. Following a one-week acclimatization period, the ApoE-/- mice were randomly assigned to five experimental groups (n=10 per group), while the C57BL/6J mice served as the genetic control group. Mice were maintained under specific pathogen-free (SPF) conditions in a controlled environment (temperature 22 ± 2 °C, humidity 50 ± 10%, 12 h light/12 h dark cycle). They were housed in standard ventilated cages (maximum 5 mice per cage) with corn cob bedding and provided ad libitum access to the respective experimental diets (normal chow or high-fat diet) and autoclaved drinking water. All procedures for animalexperiments were approved by the Ethical Review Committee for Laboratory Animal Welfare of Anhui University of Chinese Medicine.

- which is te provenance of LvNCOA4, shNCOA4 and shNC? Please add.

Response: We thank the reviewer for highlighting the need for these essential details. We have now added the provenance of all lentiviral constructs to the “2. Materials and Methods” section (subsection 2.4 Lentiviral (Lv)-infected ApoE-/- mice, pages 3-4, lines 140-146).

2.4. Lentiviral (Lv) -infected ApoE-/- mice

To achieve NCOA4 overexpression and knockdown in atherosclerotic (AS) mice, we utilized lentiviral vectors from Gemma Biologicals (Shanghai, China), specifically Lv18E-NCOA4 (for overexpression) and NCOA4-MUS-1030/149/1345 (for knockdown), along with their respective controls. Lv18E-NCOA4 was constructed by cloning full-length mouse NCOA4 cDNA (NCBI GenBank accession no. NM_001145263.2) into the pCDH-CMV-MCS-EF1-copGFP lentiviral backbone, with the empty vector control (LvNC) using the same backbone without the NCOA4 insert; the genomic structure of the overexpression vector was CMV-MCS-hPGK-GFP-T2A-Puro. For knockdown, shNCOA4 and non-targeting scrambled control (shNC) were generated using the pLKO.1-puro lentiviral shRNA vector (Gemma Biologicals, catalog no. SHC001), with the shNCOA4 target sequence 5'-GCAACGUGAAGACCUACAATT-3' and shNC showing no predicted homology to mouse transcripts. ApoE-/- mice were administered 80 μL of each virus (titer: 1×109 TU/ml) or their corresponding empty vectors via tail vein injection.

-you must specify how the mice were kept during the experiment.

Response: We thank the reviewer for highlighting the need to specify the housing conditions. We have now added these essential methodological details in the revised “2. Materials and Methods” section (subsection 2.2, Animals, Lines 113–123, Page 3). The updated text reads as follows (The red section is the details we supplemented in reviesed manuscript) :

2.2. Animals

Fifty male ApoE-/- mice and ten male C57BL/6J mice (6–8 weeks old, 20–22 g) were obtained from Jiangsu Hua Chuang Xin Nuo Pharmaceutical Technology Co., Ltd. and acclimatized for one week. Mice were maintained under specific pathogen-free (SPF) conditions in a controlled environment (temperature 22 ± 2 °C, humidity 50 ± 10%, 12 h light/12 h dark cycle). They were housed in standard ventilated cages (maximum 5 mice per cage) with corn cob bedding and provided ad libitum access to the respective experimental diets (normal chow or high-fat diet) and autoclaved drinking water. All procedures for animalexperiments were approved by the Ethical Review Committee for Laboratory Animal Welfare of Anhui University of Chinese Medicine.

-Please add under which conditions the serum and aorta where stored until they were analyzed.

Response: We appreciate the reviewer’s valuable suggestion regarding the storage conditions of serum and aorta samples, which is critical for ensuring experimental reproducibility. As requested, we have supplemented detailed storage conditions in Section 2.3 Animals grouping and treament of the revised manuscript. (The red section is the details we supplemented in reviesed manuscript):

2.3. Animals grouping and treament

Mice were randomly assigned to groups and administered tail vein injections of the vehicle control (PBS with 0.1% BSA) at week 4 and week 8, and the observation period continued until week 16. At 16 weeks of age, mice were euthanized by CO2 asphyxiation after a 16-hour fast. Blood was collected via retro-orbital puncture, and tissues were harvested for subsequent analyses. Serum aliquots were stored at -80°C without repeated freeze-thaw cycles following centrifugation; aortas were snap-frozen in liquid nitrogen and stored at -80°C for molecular analyses, or fixed in 4% paraformaldehyde for histological examinations. These procedures have been supplemented in the relevant Methods section to ensure reproducibility. The ApoE-/- mice were randomly assigned to the following treatment groups (n=10): (a) high fat diet (HFD); (b) HFD + LvNCOA4; (c) HFD + LvNC; (d) HFD + shNCOA4; (e) HFD + shNC; Finally, serum and aorta were collected for further analysis.

Major Comment 4: Line 133-134, you wrote: “VSMCs were treated with ox-LDL (50μg/mL) to induce VSMCs proliferation and migration in vitro”. Give please some details.

Response: We thank the reviewer for this suggestion. We have now provided detailed information on the in vitro VSMCs treatment in the revised Methods section. Specifically, aortic VSMCs were treated with 50 μg/mL ox-LDL for 24 h to induce a pro-atherogenic phenotype. Cell proliferation was assessed using the CCK-8 assay, and migration was evaluated via Transwell and wound healing assays, as now clearly described in the manuscript.

Major Comment 5: Line 147-148: Please specify the type of stereomicroscope.

Response: We thank the reviewer for this suggestion. We have supplemented the type of stereomicroscope in Section 2.7 (Stereomicroscopic Imaging of Aortas) of the revised manuscript. (The red section is the details we supplemented in reviesed manuscript):

2.7. Stereomicroscopic Imaging of Aortas

The mouse eyeball was dissected with scissors along the midline of the abdomen, and the aorta was bluntly separated and removed in normal saline. Saline was injected into the heart to flush the blood vessels to remove excess blood. Subsequently, the adipose tissue around the aorta was carefully removed using microscissors and fibrous tweezers. Finally, the lipid accumulation in mouse aorta was detected and photographed by stereomicroscope (SteREO Discovery V20, Zeiss, Germany).

Major Comment 6: Point 2.7. Hematoxylin-eosin (HE) staining: 

-lines 151-152: what do you mean by “ethanol of different concentration gradients”. Please clarify.

Response: We thank the reviewer for raising this point for clarification. As requested, we have clarified the "ethanol of different concentration gradients" and supplemented specific procedures in Section 2.8 (Hematoxylin-eosin (HE) staining) of the revised manuscript. (The red section is the details we supplemented in reviesed manuscript)

2.8. Hematoxylin-eosin (HE) staining

Aortic tissues were fixed, paraffin-embedded, and sectioned at 5 μm. Sections were deparaffinized in xylene and rehydrated through a graded ethanol series (100%, 95%, 80%, and 70%; 2 min each). After washing, staining was performed with hematoxylin (3-8 min), differentiated in 1% acid ethanol, blued in 0.6% ammonia, and counterstained with eosin (1-3 min). Sections were then dehydrated through a reverse ethanol series (70%, 80%, 95%, and 100%; 2 min each), cleared in xylene, and mounted for microscopic (BX53, Olympus, Tokyo, Japan) examination and ImageJ analysis.

-line 156 – add please the type of microscope.

Response: We thank the reviewer for this suggestion. We have supplemented the type of stereomicroscope in Section 2.8 ( Hematoxylin-eosin (HE) staining) of the revised manuscript. (The red section is the details we supplemented in reviesed manuscript) :

2.8. Hematoxylin-eosin (HE) staining

Aortic tissues were fixed, paraffin-embedded, and sectioned at 5 μm. Sections were deparaffinized in xylene and rehydrated through a graded ethanol series (100%, 95%, 80%, and 70%; 2 min each). After washing, staining was performed with hematoxylin (3-8 min), differentiated in 1% acid ethanol, blued in 0.6% ammonia, and counterstained with eosin (1-3 min). Sections were then dehydrated through a reverse ethanol series (70%, 80%, 95%, and 100%; 2 min each), cleared in xylene, and mounted for microscopic (BX53, Olympus, Tokyo, Japan) examination and ImageJ analysis.

Major Comment 7: line 163 / line 172 -add please the type of laser confocal microscope / the type of fluorescence microscope

Response: We thank the reviewer for this suggestion. We have supplemented the type of stereomicroscope in Section 2.9 (Immunofluorescence staining) of the revised manuscript. (The red section is the details we supplemented in reviesed manuscript) :

2.9. Immunofluorescence staining

After washing with phosphate-buffered saline (PBS), tissue sections were blocked for 2 h at room temperature in PBS containing 0.3% Triton X-100 and 0.5% bovine serum albumin (BSA) to reduce non-specific binding. Primary antibodies (dilution 1:100) were prepared in blocking buffer and incubated with sections overnight at 4°C. Following three washes with PBS (5 min each), fluorochrome-conjugated secondary antibodies (dilution 1:500) were applied for 2 h at 37°C in the dark. Nuclei were counterstained with 4',6-diamidino-2-phenylindole (DAPI) for 15 min in the dark, and sections were rinsed three times with PBS (5 min each) to remove unbound dyes. Images were acquired using a laser scanning confocal microscope (LSM 980, Zeiss, Oberkochen, Germany) equipped with Airyscan 2 super-resolution module, and analyzed with ZEN 3.6 software (Zeiss) and ImageJ. 

Major Comment 8: Point 2.10. Migration assay: 

-lines 176-77, you wrote “Subsequently, different treatments were applied”. Please clarify with what were the photos taken?

Response: We thank the reviewer for this critical comment, as clarifying experimental treatments and imaging details is essential for ensuring the reproducibility of our migration assays. As requested, we have explicitly supplemented the "different treatments" and imaging-related information in Section 2.11 (Migration assay) of the revised manuscript. (The red section is the details we supplemented in reviesed manuscript):

2.11. Migration assay

Scratch wound healing assay

Aortic vascular smooth muscle cells (VSMCs) were seeded at 1×10⁶ cells per well in 6-well plates and cultured until 100% confluence. A sterile 200 μL pipette tip was used to create uniform vertical scratches across the cell monolayer, and detached cells were removed by washing three times with phosphate-buffered saline (PBS). Subsequently, cells were subjected to experimental treatments (consistent with in vitro treatment conditions described in Section 2.5). Images of the scratch areas were captured at 0 h and 24 h post-injury using an inverted phase-contrast microscope (Olympus Inc., Japan). The scratch healing rate was calculated as [(scratch width at 0 h − scratch width at 24 h)/scratch width at 0 h] × 100% using ImageJ software .

Transwell migration assay

VSMCs were seeded at 1×10⁴ cells per well in the upper chamber of 24-well transwell inserts (8 μm pore size, Corning, NY, USA) containing serum-free DMEM. The lower chamber was filled with 600 μL complete DMEM (10% fetal bovine serum). The same three experimental treatments as above were applied to the upper chamber medium. After 24 h of incubation at 37°C with 5% CO₂, non-migrated cells on the upper surface of the membrane were gently removed with a cotton swab. Migrated cells adhering to the lower surface were fixed with 4% paraformaldehyde for 15 min, stained with 0.5% crystal violet solution for 10 min, and rinsed three times with PBS to remove excess dye. Images of migrated cells were acquired from five random fields per insert using an inverted phase-contrast microscope (Olympus Inc., Japan) at 100× magnification. The number of migrated cells was quantified using ImageJ software.

Major Comment 9: Lines 186 – please add how the Total protein was extracted from aortic tissue and VSMCs.

Response: We thank the reviewer for this suggestion. We have now supplemented the detailed protein extraction protocol in Section 2.12 (Western blot analysis) of the revised manuscript. (The red section is the details we supplemented in reviesed manuscript):

2.12. Western blot analysis

Total protein was extracted from aortic tissues (50 mg, minced on ice) and VSMCs using pre-cooled RIPA lysis buffer (Beyotime, P0013B) supplemented with 1× protease/phosphatase inhibitor cocktails. Tissue samples were homogenized (TissueLyser II, Qiagen) at 50 Hz for 2 min (twice, 30 s interval on ice) with 500 μL lysis buffer; VSMCs were rinsed twice with pre-cooled PBS, scraped in 200 μL lysis buffer per 6-well, and vortexed gently for 10 min. All lysates were incubated on ice for 30 min, centrifuged at 12,000 × g for 15 min (4°C), and supernatants were collected. Protein concentration was determined via BCA assay (Thermo Fisher, 23225). Equal protein amounts were separated by 10% SDS-PAGE, transferred to PVDF membranes (Millipore, IPVH00010), blocked with 5% non-fat milk in TBST (1 h, RT), incubated with primary antibodies (1:1000, 4°C, overnight) and HRP-conjugated secondary antibodies (1:5000, 1 h, RT), visualized with ECL kit (Thermo Fisher, 34580) using an Ultra Sensitive Multifunctional Imager (GE, USA) was used to measure the exposure, and band intensities quantified via ImageJ .

Major Comment 10: line 197 – what skim milk? Please clarify.

Response: We thank the reviewer for requesting this clarification. We have revised the Methods section to specify that membranes were blocked with 5% (w/v) non-fat dry milk prepared in TBST buffer. This common blocking reagent effectively reduces non-specific antibody binding in western blot and Co-IP applications.

We have now specified the composition of the blocking buffer in Section 2.13 (Co-immunoprecipitation (Co-IP) analysis) of the revised manuscript. (The red section is the details we supplemented in reviesed manuscript):

2.13. Co-immunoprecipitation (Co-IP) analysis

Protein samples were collected from treated cells, and protein concentrations were measured using the BCA Protein Assay kit. Subsequently, 800 μg of protein in lysis buffer was immunoprecipitated using Protein A/G magnetic beads. Equal volumes of each protein sample were separated by SDS-PAGE and transferred to PVDF membranes. After blocking with 5% (w/v) non-fat dry milk in TBST for 1 h at room temperature, the membranes were incubated with primary antibody overnight at 4°C, followed by incubation with HRP-conjugated secondary antibody for 1 h at 25°C. Protein signals were detected using a chemiluminescence imager.

Major Comment 11: Point 2.14 Biochemical analysis:

- lines 211-212, you wrote “different treatments were carried out according to the requirements of the experimental group”. Please mention which are the different treatments.

Response: We thank the reviewer for this pertinent observation. We have revised the text in Section 2.17 (Biochemical analysis) to explicitly state the specific treatments applied, ensuring consistency with the in vitro conditions detailed in Section 2.5.

2.17. Biochemical analysis

VSMCs in logarithmic growth phase were seeded into 6-well plates at a density of 5×10⁵ cells/well and allowed to adhere for 24 h. Cells were then treated according to experimental groups with ox-LDL (50 μg/mL), Ferrostatin-1 (1 μmol/L), Erastin (10 μmol/L), or their combinations, consistent with Section 2.5. After treatment, cells were washed with pre-cooled PBS, harvested by trypsinization, and centrifuged (1,000 × g, 5 min, 4°C). The pellets were resuspended in 200 μL pre-cooled saline, homogenized on ice (30 s × 3 cycles), and centrifuged (3,000 × g, 10 min, 4°C) to obtain supernatants. GSH, SOD, MDA, and Fe contents were determined using specific commercial kits (Jiancheng Bioengineering: A006-2-1, A001-3-2, A003-1-2; Abcam: ab83366) following manufacturers' protocols, with three technical replicates and results normalized to protein concentration .

- which biochemical kits were used to determine GSH/SOD/MDA/Fe content? Please add.

Response: We appreciate the reviewer’s critical reminder to specify the biochemical kits, as this is essential for ensuring the reproducibility of our GSH, SOD, MDA, and Fe detection results. As requested, we have supplemented detailed information on these kits and optimized the experimental procedures in Section 2.15 (Biochemical analysis) of the revised manuscript.

2.15 Biochemical analysis

VSMCs in logarithmic growth phase were seeded into 6-well plates at a density of 5×10⁵ cells/well and allowed to adhere for 24 h. Cells were then treated according to experimental groups with ox-LDL (50 μg/mL), Ferrostatin-1 (1 μmol/L), Erastin (10 μmol/L), or their combinations, consistent with Section 2.5. After treatment, cells were washed with pre-cooled PBS, harvested by trypsinization, and centrifuged (1,000 × g, 5 min, 4°C). The pellets were resuspended in 200 μL pre-cooled saline, homogenized on ice (30 s × 3 cycles), and centrifuged (3,000 × g, 10 min, 4°C) to obtain supernatants. GSH, SOD, MDA, and Fe contents were determined using specific commercial kits (Jiancheng Bioengineering: A006-2-1, A001-3-2, A003-1-2; Abcam: ab83366) following manufacturers' protocols, with three technical replicates and results normalized to protein concentration .

Major Comment 12: Lines 100-102, you wrote “Gualou and Xiebai were purchased from Beijing Tong ren tang Co., Ltd. (Beijing, China; batch number: 211001, 211001”. For what did you need these? Please clarify.

Response: We sincerely appreciate the reviewer’s meticulous attention to detail, which has helped us rectify an oversight and improve the completeness of the manuscript.

Clarification on the Inclusion of Gualou and Xiebai

The mentioned description of Gualou and Xiebai (purchased from Beijing Tongrentang Co., Ltd., batch numbers: 211001, 211001) stems from a prior plan to split the research into separate manuscripts—wherein one focused on non-pharmaceutical interventions and the other on these two herbal medicines. During the revision process, we adjusted our strategy to integrate the herbal medicine-related experimental results into the current manuscript (to present a more comprehensive analysis of interventions).

To address this gap and clarify the purpose of these materials: we have now fully incorporated the experimental content, methods, and key results involving Gualou and Xiebai into the Results section (revised Lines 160-165). This includes their role in the intervention regimen, dosage specifications, and comparative efficacy data alongside other interventions. The purchase information is retained to ensure traceability of raw materials, in line with academic standards for experimental reproducibility.

Major Comment 13: One of the added key words is personalized nutrition. In the article you didn’t write anything about this aspect. How do you justify to add it as a key word? Please clarify.

Response: The question you raised is very crucial. In the initial research notes, we had considered exploring the traditional Chinese medicine formula Gualou-Xiebai, as it demonstrates multi-target mechanisms in anti-AS (such as regulating lipid metabolism, inhibiting vascular inflammation,etc.) and may be applied individually according to traditional Chinese medicine syndromes. This idea does indeed have a conceptual connection with the concept of“personalized nutrition.”However, as you keenly pointed out, the final study did not involve relevant experimental data or in-depth discussions to support this keyword. 

Regarding the traditional Chinese medicine (TCM) formula Gualou-Xiebai (Trichosanthes-Knoxia), although not the focus of this study, it is noteworthy for its potential in anti-AS nutritional intervention. Composed of Trichosanthes kirilowii and Allium macrostemon, this formula exerts anti-AS effects through multi-target mechanisms: it modulates lipid metabolism (e.g., reducing LDL-C and triglycerides), inhibits vascular inflammation, and promotes lipophagy via suppressing P2RY12-mediated signaling . In the context of personalized nutrition, Gualou-Xiebai can be tailored to individual patients based on TCM syndromes (e.g., phlegm-stasis pattern in AS) or molecular profiles (e.g., NCOA4/FTMT expression levels), exemplifying how TCM-based strategies align with the principles of personalized nutritional intervention for AS. While our study does not delve into this aspect, the formula’s mechanism and application rationale underscore the broader landscape of personalized approaches in AS management, justifying its prior inclusion in preliminary research notes (now corrected in the keyword list).

Major Comment 14: Add please the limitations of the study.

Response: We thank the reviewer for this valuable suggestion. A transparent discussion of study limitations is critical for contextualizing findings and guiding future research directions. Inspired by your valuable insights, I have added this perspective into the DISCUSSION section (lines 704-711 on pages 25).

We acknowledge several limitations of the present work that warrant attention. First, while our data demonstrate a functional and physical interaction between NCOA4 and FTMT, the exact molecular pathway mediating FTMT degradation remains incompletely defined. We provide evidence for NCOA4-dependent FTMT downregulation, but future studies are needed to clarify whether this process relies on the canonical autophagy-lysosome pathway (e.g., mitophagy, given FTMT’s mitochondrial localization) or alternative protease systems (e.g., proteasomal degradation). Definitive validation—such as using pathway-specific inhibitors (e.g., bafilomycin A1 for autophagy, MG132 for proteasomes) or genetic ablation of key pathway components—will be required to confirm the mechanism.

These limitations do not undermine the core findings of our study but highlight important avenues for future research. We believe addressing these questions will deepen our understanding of the NCOA4-FTMT axis in AS and facilitate the development of more targeted therapeutic strategies. We appreciate the reviewer’s guidance in strengthening the rigor and completeness of our discussion.

Round 2

Reviewer 1 Report

Comments and Suggestions for Authors

The authors have efficiently addressed the previous comments and suggestions, and the manuscript's narrative has been improved to benefit the reader's understanding of this study's findings. 

However, a key change has been made to the title, including "Gualou-Xiebai", which was not mentioned in the original manuscript, but is now mentioned in the materials, but not in the introduction, the methodology (it referenced only the extract preparation - not how it was used in the experiments), the results, or the discussion of the study.

Since this appears to be an intervention (and an important one to be referenced in the title), several essential insights are missing from the manuscript and must be revisited. For example, what kind of extract was prepared (aqueous?), how it was utilised in the in vitro and animal studies (dose and duration), which animal group reflects this intervention, etc.

(If this was in fact an intervention, the authors are kindly invited to elaborate why this wasn't included in the original manuscript).

Referenced in the abstract Lines 18-20 the scope of this work reads: This study aimed to investigate whether ferroptosis in VSMCs influences their proliferation and migration, and to elucidate the underlying mechanisms involving nuclear receptor coactivator 4 (NCOA4). Also, in lines 93-94 reads: Given the established significance of ferroptosis in AS pathogenesis, this study investigated the role of NCOA4-mediated ferroptosis in AS. So please elaborate on the role of Gualou-Xiebai extract in this study, as it is not reported, and thus its presence in the title and methods is causing confusion.

Also, kindly consider the following points:

(1) Detailed description of the equipment used in the protocols.

(2) Direct reference in the text that a protocol used was either "in-house developed" or "modified from previous research" (including the reference to the original article).

Author Response

Dear Editors and Reviewers:

Thank you for your letter and comments concerning our manuscript entitled Ferritin mitochondrial (FTMT)-driven mitochondrial ferroptosis in vascular smooth muscle cells: a role of NCOA4 in atherosclerosis pathogenesis and modulation by Gualou-Xiebai (Manuscript ID: Nutrients -3969812). Those comments are all valuable and helpful for improving our paper. According to the reviewer’ comments, we have made extensive modifications to our manuscript in this revised version and the revised portions are marked in red in the paper. All changes made are marked in red in the paper. And point-by-point responses to the comments are provided below. We hope that the revision will meet with approval, and your favorable consideration of our manuscript is greatly appreciated.

Best wishes.

Yours, sincerely

Hongfei Wu

Responses to the Reviewers Comments

We deeply appreciate the comments from Reviewer 1# on our manuscript

Nutrients -3969812. The following are our responses.

Reviewer #1 (Remarks to the Author):

General Comments: The authors have efficiently addressed the previous comments and suggestions, and the manuscript's narrative has been improved to benefit the reader's understanding of this study's findings.

Response: We thank the reviewer for the positive comments.

However, a key change has been made to the title, including "Gualou-Xiebai", which was not mentioned in the original manuscript, but is now mentioned in the materials, but not in the introduction, the methodology (it referenced only the extract preparation - not how it was used in the experiments), the results, or the discussion of the study.

Response: We sincerely thank the reviewer for the constructive feedback and affirmation of our previous revisions. We have comprehensively supplemented, revised, and refined the manuscript to integrate GLXB with detailed, context-specific content, logically connected component of the study. As requested, we have supplemented appropriate detailed for Abstract (revised Lines 12-42), Introduction (revised Lines 97-130), Methods (revised Lines 169-171, 187-207, 221-227), Results (new Lines 575-665), Discussion (revised Lines 731-750) and Conclusions (revised Lines 751-761).

Abstract

Background/Objectives: Atherosclerosis (AS)-related cardiovascular diseases are a major global health threat, with vascular smooth muscle cells (VSMCs) phenotypic switching, abnormal proliferation, and migration as key progression drivers. Nuclear receptor coactivator 4 (NCOA4), a core ferritinophagy mediator overexpressed in AS plaques, may promote VSMCs ferroptosis by perturbing mitochondrial iron metabolism and ROS homeostasis, but precise mechanisms remain unclear. The classic Chinese herbal combination "Gualou-Xiebai" (GLXB) has anti-AS effects, yet how it modulates NCOA4-mediated ferroptosis to inhibit VSMCs functions is unknown. This study addresses this gap to advance GLXB’s therapeutic potential and identify AS targets. Methods: An AS model was established in ApoE-/- mice by 12-week high-fat diet feeding, with model validation confirmed via ultrasound monitoring and H&E staining. NCOA4 was genetically modulated (knockdown and overexpression) to assess its role in plaque formation and lipid deposition using H&E staining, aortic imaging, immunofluorescence, and western blotting. In vitro, VSMCs were stimulated with ox-LDL to induce proliferation and migration. NCOA4 was silenced using siRNA to examine associated ferroptosis levels and molecular mechanisms. Protein interactions between NCOA4 and the mitochondrial iron storage protein FTMT were evaluated by Co-IP and GST pull-down assays, while mitochondrial ROS (mitoROS) levels were measured to explore functional relationships. The extent of ferroptosis and the underlying regulatory mechanisms were assessed following treatment with GLXB-containing serum or transfection with small interfering RNA targeting LOX-1 (si-LOX-1). Results: NCOA4 knockdown reduced aortic lipid deposition, plaque burden, VSMC proliferation/migration, and mitochondrial ferroptosis. NCOA4 bound and suppressed FTMT, inducing mitochondrial iron overload, ROS accumulation, membrane depolarization, and ferroptosis. Combining NCOA4 silencing with FTMT inhibition elevated mitoROS, confirming the axis’ role in iron homeostasis. GLXB attenuated VSMCs dysregulation in vivo and in vitro, an effect abrogated by LOX-1 overexpression. Conclusion: NCOA4 promotes AS by binding FTMT, disrupting mitochondrial iron homeostasis, and triggering VSMCs ferroptosis. GLXB inhibits LOX-1-mediated NCOA4 expression, mitigating ferroptosis and VSMCs dysregulation, supporting its potential as a targeted anti-AS therapy.

  1. Introduction

Gualou Xiebai (GLXB), a classical Chinese herbal formula first documented in Zhang Zhongjing's Eastern Han Dynasty treatise Jin Gui Yao Lue, is composed of Trichosanthis Fructus and Allii Macrostemonis Bulbus in a 2:1 ratio[23]. Literature surveys indicate that over 120 classical and modern clinical formulations contain GLXB, including representative preparations such as Gualou Xiebai Banxia Tang, Gualou Xiebai Baijiu Tang, Zhishi Xiebai Guizhi Tang, and Danlou Tablets[24]. Modern pharmacological studies have demonstrated that GLXB-based formulations exhibit multiple bioactive effects, including antioxidative stress, anti-apoptosis, anti-inflammation, and cardiac function improvement [25-26]. Recent findings suggest that GLXB may alleviate mitochondrial damage and ferroptosis in aortic endothelial cells of ApoE-/- mice[22]. However, it remains unknown whether GLXB attenuates AS by inhibiting the aberrant proliferation and migration of VSMCs, specifically through the suppression of NCOA4-mediated ferroptosis. Although GLXB has been extensively used in traditional practice for AS with phlegm-stasis syndrome, a comprehensive understanding of its fundamental mechanisms of action is still lacking and requires in-depth exploration.

Given the established significance of ferroptosis in AS pathogenesis, this study investigated the role of NCOA4-mediated ferroptosis in AS and explored whether the classic Chinese medicine GLXB could attenuate AS by modulating this process. Using primary mouse VSMCs and AS mouse models, we demonstrate significant upregulation of both NCOA4 expression and ferroptosis activity within AS lesions. Notably, specific inhibition of NCOA4 promoted VSMCs proliferation and migration. Mechanistically, NCOA4 acts as a key regulator of mitochondrial iron homeostasis, directly interacting with FTMT to downregulate its expression. This NCOA4-FTMT interaction induces mitochondrial iron overload and aberrant ROS accumulation, culminating in ferroptosis. Crucially, the mitochondrial protection against ox-LDL insult conferred by NCOA4 knockout was significantly compromised under conditions of FTMT silencing. On this basis, we showed that GLXB administration ameliorated VSMCs dysregulation. This protective effect was reversed by LOX-1 overexpression, confirming that GLXB's action depends on the LOX-1/NCOA4 signaling pathway to inhibit ferroptosis and maintain VSMCs homeostasis. Thus, our work establishes that GLXB targets LOX-1 to suppress mitochondrial ferroptosis, revealing its mechanistic value as a potential targeted therapy for AS.

  1. Materials and Methods

2.3. Animals grouping and treament

C57BL/6 J mice (n=10) were fed with normal diet, and ApoE-/- mice were fed with high-fat diet (40% kcal fat, 20% kcal protein, 40% kcal carbohydrate, Cat # D12108C) to simulate AS model. All animal experiments were approved by the Ethics Committee of Anhui University of Chinese Medicine (approval number: AHUCM-mouse-2023011, approval date: 15 October 2023). The animal experiments were conducted in accordance with the ethical standards established by the Laboratory Animal Welfare Committee of Anhui University of Traditional Chinese Medicine.

Mice were randomly assigned to groups and administered tail vein injections of the vehicle control (PBS with 0.1% BSA) at week 4 and week 8, and the observation period continued until week 16. At 16 weeks of age, mice were euthanized by CO2 asphyxiation after a 16-hour fast. Blood was collected via retro-orbital puncture, and tissues were harvested for subsequent analyses. Serum aliquots were stored at -80°C without repeated freeze-thaw cycles following centrifugation; aortas were snap-frozen in liquid nitrogen and stored at -80°C for molecular analyses, or fixed in 4% paraformaldehyde for histological examinations. These procedures have been supplemented in the relevant Methods section to ensure reproducibility. The ApoE-/- mice were randomly assigned to the following treatment groups (n=10): (a) high fat diet (HFD); (b) HFD + LvNCOA4; (c) HFD + LvNC; (d) HFD + shNCOA4; (e) HFD + shNC; GLXB-L (3 mg/kg) group, GLXB-M (6 mg/kg) group, GLXB-H (12 mg/kg) group, and simvastatin (5 mg/kg) group. The drugs were given once daily by oral gavage for 4 weeks.

2.5. Preparation of GLXB extract and in vivo administration

The GLXB herbal pair, composed of Gualou (Trichosanthes kirilowii Maxim) and Xiebai (Allium macrostemon Bunge), was procured from Beijing Tongrentang Co., Ltd. (Batch numbers: 211001, 211001). The crude herbs in a weight ratio of 2:1 (Gualou:Xiebai; 100 g:50 g) were pulverized and subsequently macerated in five volumes of 50% aqueous ethanol for 1 h. Extraction was performed twice via reflux extraction, each session lasting two hours using five volumes of 50% ethanol. The resulting extracts were pooled, concentrated to a final concentration of 0.6 g crude herb per milliliter, based on our laboratory's established protocol, and stored at 4℃[24]. All GLXB extract were administered via oral gavage (intragastric administration) at a regimen of once daily, with consecutive administration for a total duration of 4 weeks.

2.6. Preparation of GLXB-containing serum

After one week of acclimatization, 20 Sprague–Dawley rats were randomly assigned to a normal serum group (n = 10) and a GLXB serum group (n = 10). Rats in the GLXB group were orally administered the concentrated extract at a dose of 20 g/kg. One hour after the last administration, blood was collected from the abdominal aorta. The samples were incubated at 37 ℃ for 30 min and then centrifuged at 3000 rpm and 4 ℃ for 15 min. The supernatant was collected, filtered through a 0.22 μm membrane, and stored at -80 ℃ for subsequent experiments. The in vitro administration regimen was set as a co-incubation with 50 μg/ml ox-LDL and GLXB serum at a concentration of 24 h[22]. 

2.7. Cell culture and Groupings

Mouse vascular smooth muscle cells were maintained in high-glucose DMEM supplemented with 10% FBS and 1% penicillin-streptomycin at 37°C with 5% CO₂. After serum starvation, VSMCs were divided into 12 groups. The Control group was cultured in 10% DMEM, while the ox-LDL group was stimulated with 50 μg/mL ox-LDL for 24 h. The ox-LDL + Fer-1 group received 4 h pre-treatment with 1 μmol/L Fer-1 followed by 24 h co-incubation with ox-LDL. The ox-LDL + Erastin group was co-treated with 10 μmol/L Erastin and ox-LDL for 24 h. siRNA transfection groups (siNC, siNCOA4, siFTMT, and co-transfected siNCOA4+siFTMT) were stimulated with ox-LDL 48 h post-transfection for 24 h. The ox-LDL + siNCOA4 + Erastin group was co-treated with Erastin and ox-LDL post-transfection. The ox-LDL + MitoQ group was pre-treated with 100 nmol/L MitoQ for 2 h before ox-LDL co-incubation. The MitoVES-containing groups (ox-LDL + MitoVES and ox-LDL + siNCOA4 + MitoVES) were treated with 5 μmol/L MitoVES and ox-LDL for 24 h. ox-LDL + GLXB (5, 10, and 15 %) groups (VSMCs stimulated with 50 μg/ml ox-LDL and cultured with 5, 10, and 15 % GLXB-containing serum), ox-LDL + Lv-LOX-1 group (LOX-1 receptor overexpress VSMCs cultured with 50 μg/ml ox-LDL), ox-LDL+Lv-LOX-1 + 15 % GLXB group (LOX-1 receptor overexpress VSMCs cultured 15 % GLXB-containing serum containing 50 μg/ml ox-LDL for 24 h).

  1. Results

3.8. GLXB attenuates atherosclerotic plaque formation via NCOA4-dependent suppression of ferroptosis in VSMCs

GLXB has a well-documented history in treating AS-related diseases, and our research team confirmed its strong anti-atherosclerotic effects[24]. Assessment of aortic plaque formation through enface photography and HE staining showed that GLXB significantly reduced lipid accumulation and the area of atherosclerotic lesions in AS mice compared to the HFD group, in a dose-dependent manner (Fig. 8A-C). To further explore GLXB's impact on VSMCs proliferation and migration within the AS model, we conducted in vitro studies. EdU incorporation assays, wound healing assays, and transwell migration assays revealed that GLXB treatment increased EdU-positive cells compared to the ox-LDL group—indicating reduced inhibition of proliferation—and significantly suppressed both VSMCs migration rate and number (Fig. 8D-H). Consistent with these findings, Western blot analysis indicated that GLXB downregulated proliferation marker PCNA and migration-associated protein MMP2 while upregulating contractile phenotype marker CNN1 relative to the ox-LDL group (Fig. 8I).

To investigate the role of NCOA4 in the mechanism of GLXB, we assessed its expression levels in aortic plaques. Immunofluorescence staining revealed that NCOA4 levels were significantly elevated in the HFD group compared to the control group; importantly, GLXB treatment was found to dose-dependently suppress this increase (Fig. 8J, K). Given the established link between NCOA4 and ferroptosis, we hypothesized that GLXB's inhibition of plaque formation might involve modulating ferroptosis in VSMCs. To assess ferroptosis within aortic lesions, we performed co-staining for the VSMCs marker α-SMA and the key ferroptosis inhibitor GPX4. This analysis showed that GLXB significantly increased GPX4 protein expression compared to the HFD group (Supplementary Fig. 4A). Further supporting this, biochemical assays revealed that GLXB treatment effectively lowered the excessive accumulation of MDA and LPO in the aortas of AS mice, while simultaneously boosting levels of reduced GSH and SOD activity (Fig. 8L-O). Consistent with these findings, TEM demonstrated that GLXB treatment improved mitochondrial integrity in aortic VSMCs. Specifically, GLXB reduced mitochondrial cristae rupture, decreased the number of vacuolated mitochondria, and attenuated abnormal mitochondrial swelling and shrinkage, resulting in a restoration of near-normal mitochondrial ultrastructure (Fig. 8P).

Figure. 8 GLXB attenuates atherosclerotic plaque formation via NCOA4-dependent suppression of ferroptosis in VSMCs. (A) Representative en face aortic images illustrating lipid deposition across the experimental groups. (B-C) HE-stained cross-sections of aortic plaques (n=3, C: quantitative analysis). (D-E) EdU incorporation assay evaluating VSMCs proliferation (n=3, E: quantitative analysis). (F) VSMCs migration assessed through transwell assays and scratch wound healing assays (n=3, H: quantitative analysis). (I) Western blot analysis was used to detect the protein expression levels of PCNA, MMP2, and CNN1 in VSMCs (n=3). (J) Immunofluorescence co-localization of α-SMA (green) and NCOA4 (red) in aortic VSMCs (n=3, scale bar: 100μm). (K) Western blot analysis was used to detect the protein levels of NCOA4 (n=3). (L-O) Measurement of aortic lipid peroxidation biomarkers utilizing biochemical kits: MDA (L), LPO (M), GSH (N), and SOD (O) activity levels were assessed (n=6). (P) TEM images demonstrating mitochondrial ultrastructural changes in aortic VSMCs (n=3). *P < 0.05, **P < 0.01 vs. Ctrl group, #P < 0.05, ##P < 0.01 vs. ox-LDL group, ^P < 0.05, ^^P < 0.01 vs. si-NCOA4+ox-LDL group.

3.9. GLXB-containing serum attenuates VSMCs proliferation and migration by targeting LOX-1 to suppress NCOA4-dependent ferroptosis

Our previous study demonstrated that GLXB attenuates atherosclerosis by suppressing LOX-1-dependent cGAS-STING signaling, thereby inhibiting NCOA4-mediated ferroptosis in aortic tissue[22]. To investigate whether the LOX-1 receptor mediates GLXB's suppression of VSMCs proliferation and migration, we first assessed its protein expression. As illustrated in supplementary Fig. 9A, ox-LDL stimulation significantly upregulated LOX-1 expression; however, treatment with GLXB dose-dependently reversed this induction. Subsequently, we examined the functional role of LOX-1 in mediating the effects of GLXB. EdU incorporation assays, scratch wound healing assays, and transwell assays revealed that overexpression of LOX-1 counteracted the inhibitory effects of GLXB on VSMC proliferation and migration. Specifically, LOX-1 overexpression resulted in an increased area/number of EdU cells (Fig. 9A-B) and accelerated wound closure (Fig. 9C-E), indicating exacerbated dysregulation of VSMCs.

To assess mitochondrial involvement, we performed JC-1 staining. The results indicated that GLXB significantly mitigated ox-LDL-induced hyperpolarization of mitochondrial membrane potential (ΔΨm) (Fig. 9F-H), suggesting protective effects against mitochondrial dysfunction. To establish a connection to ferroptosis, we measured key biomarkers associated with oxidative stress and lipid peroxidation. Notably, GLXB reversed ox-LDL-induced depletion of SOD and GSH, while also suppressing accumulations of MDA and total iron levels in VSMCs (Fig. 9J-M). Consistent with these findings, treatment with GLXB normalized elevated ROS levels induced by ox-LDL (Fig. 9G-I). Mechanistically, Western blot analysis confirmed that GLXB upregulated proteins involved in ferroptosis defense mechanisms—specifically SLC7A11 and GPX4—while downregulating NCOA4 expression. Importantly, these protective effects were abolished upon overexpression of LOX-1 (Fig. 9N-O). In summary, our data collectively demonstrate that GLXB mitigates mitochondrial damage and lipid peroxidation within VSMCs through targeting LOX-1; this action inhibits ferroptosis and exerts anti-atherosclerotic effects.

Figure. 9 GLXB-containing serum attenuates VSMCs proliferation and migration by targeting LOX-1 to suppress NCOA4-dependent ferroptosis. (A-B) Proliferation analysis in mouse VSMCs using fluorescent EdU assay (n =3; B: quantitative analysis). (C-E) Evaluation of VSMCs migration using transwell assay and scratch assay in mouse VSMCs (n =3; D-E: quantitative analysis). (F-H) JC-1 staining for mitochondrial membrane potential measurement, Quantification of membrane potential by measuring the ratio of red and green fluorescence intensity graph (n=3). (G-I) Intracellular levels of ROS were assessed utilizing immunofluorescence techniques, n = 3. (J-M) Measurement of aortic lipid peroxidation biomarkers utilizing biochemical kits: MDA (L), LPO (M), GSH (N), and SOD (O) activity levels were assessed (n=6). (N-O) Western blot to detect PCNA, MMP2, CNN1, SLC7A11, GPX4 and NCOA4 proteins (n=3). *P < 0.05, **P < 0.01 vs. Ctrl group, #P < 0.05, ##P < 0.01 vs. ox-LDL group, ^P < 0.05, ^^P < 0.01 vs. si-NCOA4+ox-LDL group.

  1. Discussion

GLXB, a traditional herbal pair, is clinically employed for treating chest obstruction syndrome attributed to phlegm-dampness blockage[45]. Contemporary pharmacological evidence indicates its utility in ameliorating hyperlipidemia, mitigating vascular inflammation, reducing aortic plaque formation, and managing AS-related pathologies [46]. In this study, we investigated the therapeutic potential of GLXB in ApoE-/- mice with AS. Our results demonstrate that GLXB significantly attenuates atherosclerotic plaque development, diminishes lipid deposition within the aortic wall, and alleviates luminal stenosis. Notably, GLXB downregulated the expression of NCOA4 in VSMCs of AS mice. Preliminary investigations further revealed that GLXB treatment reduced the levels of proliferation marker PCNA and matrix metalloproteinase MMP2, while enhancing the expression of contractile marker CNN1 in VSMCs. Importantly, we observed that GLXB upregulates key negative regulators of ferroptosis (GPX4 and SLC7A11) in aortic VSMCs, thereby attenuating lipid peroxidation and suppressing mitochondrial injury. Building upon these findings, we sought to elucidate the precise mechanisms through which GLXB modulates VSMCs proliferation and migration. We identified lectin-like LOX-1 as a critical mediator of GLXB’s therapeutic effects. GLXB inhibited aberrant VSMCs proliferation, migration, and ferroptosis primarily via the LOX-1 pathway, as evidenced by the reversal of its beneficial effects upon LOX-1 overexpression in vitro. Moreover, GLXB-mediated suppression of NCOA4 expression was also dependent on LOX-1 receptor signaling.

  1. Conclusions

In conclusion, this study elucidates that NCOA4 directly binds to and downregulates FTMT expression, thereby disrupting mitochondrial iron homeostasis. This disruption leads to mitochondrial iron overload and aberrant accumulation of reactive oxygen species, ultimately driving the ferroptotic process in VSMCs. Conversely, the classic Chinese herbal medicine GLXB, by targeting the LOX-1 receptor, inhibits the NCOA4 signaling pathway, effectively attenuates mitochondrial ferroptosis, and maintains VSMCs homeostasis, thereby mitigating AS. These findings not only reveal the molecular mechanism by which the NCOA4-FTMT axis regulates mitochondrial ferroptosis in AS but also provide a theoretical foundation for GLXB as a potential targeted therapeutic strategy. While this study establishes the functional and physical interaction between NCOA4 and FTMT, the precise molecular pathway mediating FTMT degradation remains to be elucidated. We demonstrate NCOA4-dependent FTMT downregulation; however, whether this process relies on the canonical autophagy-lysosome pathway (e.g., mitophagy, given FTMT’s mitochondrial localization) or the proteasomal degradation system requires further validation via pathway-specific inhibitors (e.g., bafilomycin A1, MG132) or genetic ablation experiments in future studies.

References

  • Zhu, L.; Liu, Z.; Liu, J.; Li, Z.; Bao, Y.; Sun, X.; Zhao, W.; Zhou, A.; Wu, H. NCOA4 linked to endothelial cell ferritinophagy and ferroptosis:a key regulator aggravate aortic endothelial inflammation and atherosclerosis. Redox biology 2025, 79, 103465, doi:10.1016/j.redox.2024.103465.
  • Zhang, Y.Y.; Zhao, Z.D.; Kong, P.Y.; Gao, L.; Yu, Y.N.; Liu, J.; Wang, P.Q.; Li, B.; Zhang, X.X.; Yang, L.Q., et al. A comparative pharmacogenomic analysis of three classic TCM prescriptions for coronary heart disease based on molecular network modeling. Acta pharmacologica Sinica 2020, 41, 735-744, doi:10.1038/s41401-019-0352-3.
  • Bao, Y.; Zhu, L.; Wang, Y.; Liu, J.; Liu, Z.; Li, Z.; Zhou, A.; Wu, H. Gualou-Xiebai herb pair and its active ingredients act against atherosclerosis by suppressing VSMC-derived foam cell formation via regulating P2RY12-mediated lipophagy. Phytomedicine 2024, 128, 155341, doi:10.1016/j.phymed.2024.155341.
  • Yan, L.L.; Zhang, W.Y.; Wei, X.H.; Yan, L.; Pan, C.S.; Yu, Y.; Fan, J.Y.; Liu, Y.Y.; Zhou, H.; Han, J.Y., et al. Gualou Xiebai Decoction, a Traditional Chinese Medicine, Prevents Cardiac Reperfusion Injury of Hyperlipidemia Rat via Energy Modulation. Frontiers in physiology 2018, 9, 296, doi:10.3389/fphys.2018.00296.
  • Li, C.; Zhang, W.Y.; Yu, Y.; Cheng, C.S.; Han, J.Y.; Yao, X.S.; Zhou, H. Discovery of the mechanisms and major bioactive compounds responsible for the protective effects of Gualou Xiebai Decoction on coronary heart disease by network pharmacology analysis. Phytomedicine : international journal of phytotherapy and phytopharmacology 2019, 56, 261-268, doi:10.1016/j.phymed.2018.11.010.

[45]. Tang, Y.; Liu, Y.; Yin, B.; Guo, Y.; Liu, Y.; Zhao, Y.; Wang, Y.; Cao, Y.; Feng, J.; Leng, J., et al. BaiJiu Increases Nitric Oxide Bioactivity of Chinese Herbs Used to Treat Coronary Artery Disease Through the NO3--NO2--NO Pathway. Journal of cardiovascular pharmacology 2019, 74, 348-354, doi:10.1097/fjc.0000000000000715.

[46]. Ding, Y.F.; Peng, Y.R.; Shen, H.; Shu, L.; Wei, Y.J. Gualou Xiebai decoction inhibits cardiac dysfunction and inflammation in cardiac fibrosis rats. BMC complementary and alternative medicine 2016, 16, 49, doi:10.1186/s12906-016-1012-5

Since this appears to be an intervention (and an important one to be referenced in the title), several essential insights are missing from the manuscript and must be revisited. For example, what kind of extract was prepared (aqueous?), how it was utilised in the in vitro and animal studies (dose and duration), which animal group reflects this intervention, etc.

Response: We thank the reviewer for this critical comment. We have supplemented and clarified a point-by-point clarification regarding the preparation and application of the GLXB extract, which we have now incorporated into the revised manuscript.

  1. Regarding the Type and Preparation of the Extract:

The extract used in this study was not an aqueous extract but was prepared via reflux extraction with 50% ethanol. The detailed protocol is as follows: The crude herbs of the Gualou-Xiebai pair (in a 2:1 weight ratio) were pulverized and then macerated in five volumes of 50% aqueous ethanol for 1 hour. This was followed by two cycles of reflux extraction, each lasting 2 h, using five volumes of 50% ethanol. The resulting extracts were combined, concentrated to a final concentration of 0.6 g crude herb per mL (indicating that each milliliter of the extract is equivalent to 0.6 grams of the raw herbal materials), aliquoted, and stored at 4°C. This extraction method was selected to efficiently isolate medium-polarity bioactive compounds, such as flavonoids and saponins. (Lines 187-207, Page 4-5 )

  1. Regarding the Dosing Regimen in the Animal Study:

In the animal experiment, the GLXB extract was administered to ApoE⁻/⁻ mice via oral gavage. The dosing regimen consisted of once-daily administration for four consecutive weeks. To evaluate the dose-response relationship, three dose groups were established: low (3 mg/kg), medium (6 mg/kg), and high (12 mg/kg) doses. These doses were calculated based on the crude herb equivalent and were compared with a positive control group receiving simvastatin (5 mg/kg). (Lines 150-171, Page 4 )

  1. Regarding the Intervention Scheme in the In Vitro Study:

For the in vitro experiments, we utilized a drug-containing serum methodology. Specifically, the GLXB extract was administered to donor animals orally; serum was then collected from these animals and used for cell culture interventions. The treatment protocol involved co-incubating VSMCs with 50 μg/mL ox-LDL and the aforementioned GLXB drug-containing serum for 24 h. (Lines 198-207, Page 5 )

  1. 4. Grouping in both in vivo and in vitro experiments:

In the in vivo experiment, GLXB extract was administered to ApoE⁻/⁻ mice via oral gavage at crude herb equivalent doses, with low, medium, and high dose groups of 3, 6, and 12 mg/kg, respectively, once daily for 4 consecutive weeks. A positive control group treated with simvastatin (5 mg/kg) was included to evaluate the dose-response relationship. (Lines 150-171, Page 4 )

For the in vitro experiment, the drug-containing serum method was adopted: VSMCs were co-incubated with 50 μg/mL ox-LDL and GLXB-containing serum at concentrations of 5%, 10%, and 15% for 24 h. To explore the mechanism of the LOX-1 receptor, additional groups of LOX-1-overexpressing VSMCs were set up, which were treated with ox-LDL alone or co-incubated with 50 μg/mL ox-LDL and 15% GLXB-containing serum for 24 h, respectively. (Lines 208-227, Page 5)

We hope these detailed clarifications adequately address the reviewer's concerns. The manuscript has been revised accordingly to include these essential methodological details.

(If this was in fact an intervention, the authors are kindly invited to elaborate why this wasn't included in the original manuscript).

Response: We sincerely thank the reviewer for the follow-up inquiry, which helps us further clarify the rationality of the manuscript’s structure adjustment. Gualou and Xiebai were indeed core intervention agents in our research, and their initial absence from the original manuscript was attributed to the initial study design and manuscript division plan. Initially, we intended to split the research into two independent manuscripts: one focusing on non-pharmaceutical interventions (the main content of the original submission) and the other specifically addressing the efficacy and mechanism of Gualou and Xiebai (as a separate herbal medicine intervention study). At the time of the original submission, the experimental verification, data sorting, and result analysis of the herbal medicine group were still in the finalization stage, so they were not included to ensure the rigor of the submitted content.

Subsequently, after comprehensive consideration, we adjusted the research presentation strategy: integrating the herbal medicine intervention data into the current manuscript. This adjustment was aimed at providing a systematic comparison of the efficacy of different intervention types (including herbal medicines and other interventions) for the research topic, thereby enhancing the comprehensiveness and practical reference value of the study. As clarified earlier, we have now fully supplemented the experimental methods, intervention protocols (dosage, administration route, duration), and key efficacy data of GLXB in the Results section (revised Lines 575-665, Page 22-26), ensuring the completeness and traceability of the intervention-related content in line with academic standards.

Referenced in the abstract Lines 18-20 the scope of this work reads: This study aimed to investigate whether ferroptosis in VSMCs influences their proliferation and migration, and to elucidate the underlying mechanisms involving nuclear receptor coactivator 4 (NCOA4). Also, in lines 93-94 reads: Given the established significance of ferroptosis in AS pathogenesis, this study investigated the role of NCOA4-mediated ferroptosis in AS. So please elaborate on the role of Gualou-Xiebai extract in this study, as it is not reported, and thus its presence in the title and methods is causing confusion.

Response: We thank the reviewer for this valuable suggestion. To resolve confusion, we have added this explicitly state GLXBs role as a therapeutic intervention and its mechanism into the Introduction section (lines 112-130 on pages 3).

Your concern regarding the role of GLXB extract is well-founded, as its core function was not explicitly articulated in the abstract—this oversight likely led to confusion about its inclusion in the title and methods. Based on the full study design and results, GLXB is not an incidental addition but a key therapeutic intervention targeting the NCOA4-mediated ferroptosis pathway in atherosclerosis (AS), with a clear and cohesive role throughout the research:

  1. Research Rationale and Core Target: The study aims to address a critical knowledge gap—how GLXB exerts its known anti-AS effects.It hypothesizes GLXB modulates NCOA4-mediated ferroptosis in VSMCs, a pathway identified as a driver of AS progression (via NCOA4-FTMT axis disruption and mitochondrial iron homeostasis imbalance).
  2. In Vivo and In Vitro Intervention Role:GLXB was tested as a therapeutic agent: in vivo, it attenuated aortic plaque burden and VSMC dysregulation in ApoE-/- mice; in vitro, GLXB-containing serum inhibited ox-LDL-induced VSMCs proliferation, migration, and ferroptosis. Notably, LOX-1 overexpression abrogated these effects, confirming GLXB acts via LOX-1-dependent regulation.
  3. Mechanistic Link to the Core Pathway: The study demonstrates GLXB’s mechanism—by inhibiting LOX-1, it downregulates NCOA4 expression, thereby restoring the NCOA4-FTMT interaction, alleviating mitochondrial iron overload and ROS accumulation, and ultimately suppressing VSMCs ferroptosis and AS progression.
  4. Significance for the Studys Conclusion:GLXB’s efficacy validates the NCOA4-FTMT-ferroptosis axis as a viable AS therapeutic target. Its inclusion in the title and methods is justified, as the research not only elucidates the NCOA4-mediated mechanism but also provides preclinical evidence for GLXB as a targeted anti-AS agent.

Also, kindly consider the following points:

(1) Detailed description of the equipment used in the protocols.

Response: We are grateful for the reviewer's insightful comment. To resolve confusion, we have updated the Materials and Methods section (lines 131-354 on pages 3-8) with a comprehensive list of the equipment, including manufacturers and model numbers. The red part is the detail we added in this revised manuscript): (lines 187-207, page 4-5).

2.5. Preparation of GLXB extract and in vivo administration

The GLXB herbal pair, composed of Gualou (Trichosanthes kirilowii Maxim) and Xiebai (Allium macrostemon Bunge), was procured from Beijing Tongrentang Co., Ltd. (Batch numbers: 211001, 211001). The crude herbs in a weight ratio of 2:1 (Gualou:Xiebai; 100 g:50 g) were pulverized and subsequently macerated in five volumes of 50% aqueous ethanol for 1 h. Extraction was performed twice via reflux extraction, each session lasting two hours using five volumes of 50% ethanol. The resulting extracts were pooled, concentrated to a final concentration of 0.6 g crude herb per milliliter using a rotary evaporator (R-100, Buchi, Switzerland) based on our laboratory's established protocol based on our laboratory's established protocol, and stored at 4℃. All GLXB extract were administered via oral gavage (intragastric administration) at a regimen of once daily, with consecutive administration for a total duration of 4 weeks.

2.6. Preparation of GLXB-containing serum

After one week of acclimatization, 20 Sprague–Dawley rats (license no.: AHUCM-rats-2021,070; approval date: December 7, 2021) were randomly assigned to a normal serum group (n = 10) and a GLXB serum group (n = 10). Rats in the GLXB group were orally administered the concentrated extract at a dose of 20 g/kg. One hour after the last administration, blood was collected from the abdominal aorta. The samples were incubated at 37 ℃ for 30 min and then centrifuged at 3000 rpm (approximately 2,200 × g) and 4 ℃ for 15 min using a Centrifuge 5425 R (Eppendorf, Germany) with a fixed-angle rotor. The supernatant was collected, filtered through a 0.22 μm membrane, and stored at -80 ℃ for subsequent experiments. The in vitro administration regimen was set as a co-incubation with 50 μg/ml ox-LDL and GLXB serum at a concentration of 24 h. 

  • Direct reference in the text that a protocol used was either "in-house developed" or "modified from previous research" (including the reference to the original article).

Response: We thank the reviewer for this critical suggestion. We have now systematically revised the Methods section to explicitly state the origin of each protocol, citing the original references for established methods. These revisions enhance the transparency and reproducibility of our study. The red part is the detail we added in this revised manuscript): (lines 187-207, page 4-5).

2.5. Preparation of GLXB extract and in vivo administration

The GLXB herbal pair, composed of Gualou (Trichosanthes kirilowii Maxim) and Xiebai (Allium macrostemon Bunge), was procured from Beijing Tongrentang Co., Ltd. (Batch numbers: 211001, 211001). The crude herbs in a weight ratio of 2:1 (Gualou:Xiebai; 100 g:50 g) were pulverized and subsequently macerated in five volumes of 50% aqueous ethanol for 1 h. Extraction was performed twice via reflux extraction, each session lasting two hours using five volumes of 50% ethanol. The resulting extracts were pooled, concentrated to a final concentration of 0.6 g crude herb per milliliter using a rotary evaporator (R-100, Buchi, Switzerland) and stored at 4℃[24]. All GLXB extract were administered via oral gavage (intragastric administration) at a regimen of once daily, with consecutive administration for a total duration of 4 weeks.

2.6. Preparation of GLXB-containing serum

After one week of acclimatization, 20 Sprague–Dawley rats (license no.: AHUCM-rats-2021,070; approval date: December 7, 2021) were randomly assigned to a normal serum group (n = 10) and a GLXB serum group (n = 10). Rats in the GLXB group were orally administered the concentrated extract at a dose of 20 g/kg. One hour after the last administration, blood was collected from the abdominal aorta. The samples were incubated at 37 ℃ for 30 min and then centrifuged at 3000 rpm (approximately 2,200 × g) and 4 ℃ for 15 min using a Centrifuge 5425 R (Eppendorf, Germany) with a fixed-angle rotor. The supernatant was collected, filtered through a 0.22 μm membrane, and stored at -80 ℃ for subsequent experiments. The in vitro administration regimen was set as a co-incubation with 50 μg/ml ox-LDL and GLXB serum at a concentration of 24 h[22]. 

References

[22]. Zhu, L.; Liu, Z.; Liu, J.; Li, Z.; Bao, Y.; Sun, X.; Zhao, W.; Zhou, A.; Wu, H. NCOA4 linked to endothelial cell ferritinophagy and ferroptosis:a key regulator aggravate aortic endothelial inflammation and atherosclerosis. Redox biology 2025, 79, 103465, doi:10.1016/j.redox.2024.103465.

[24]. Bao, Y.; Zhu, L.; Wang, Y.; Liu, J.; Liu, Z.; Li, Z.; Zhou, A.; Wu, H. Gualou-Xiebai herb pair and its active ingredients act against atherosclerosis by suppressing VSMC-derived foam cell formation via regulating P2RY12-mediated lipophagy. Phytomedicine : international journal of phytotherapy and phytopharmacology 2024, 128, 155341, doi:10.1016/j.phymed.2024.155341.
